# Learning to Condition: A Neural Heuristic for Scalable MPE Inference

**Brij Malhotra**
Department of Computer Science
The University of Texas at Dallas
brijgulsharan.malhotra@utdallas.edu

**Shivvrat Arya**
Department of Computer Science
New Jersey Institute of Technology
shivvrat.arya@njit.edu

**Tahrima Rahman**
Department of Computer Science
The University of Texas at Dallas
tahrima.rahman@utdallas.edu

**Vibhav Gogate**
Department of Computer Science
The University of Texas at Dallas
vibhav.gogate@utdallas.edu

## Abstract

We introduce *learning to condition* (L2C), a scalable, data-driven framework for accelerating Most Probable Explanation (MPE) inference in Probabilistic Graphical Models (PGMs), a fundamentally intractable problem. L2C trains a neural network to score variable-value assignments based on their utility for conditioning, given observed evidence. To facilitate supervised learning, we develop a scalable data generation pipeline that extracts training signals from the search traces of existing MPE solvers. The trained network serves as a heuristic that integrates with search algorithms, acting as a conditioning strategy prior to exact inference or as a branching and node selection policy within branch-and-bound solvers. We evaluate L2C on challenging MPE queries involving high-treewidth PGMs. Experiments show that our learned heuristic significantly reduces the search space while maintaining or improving solution quality over state-of-the-art methods.

## 1 Introduction

Probabilistic Graphical Models (PGMs) [1], such as Bayesian Networks and Markov Networks, efficiently encode joint probability distributions over a large set of random variables, enabling structured reasoning under uncertainty. A fundamental inference task in these models is the Most Probable Explanation (MPE) query, where the goal is to find the most likely assignment of values to unobserved variables given observed evidence.

Answering MPE queries is computationally intractable in general due to their NP-hardness, and becomes particularly challenging as model complexity and scale increase. Classical exact methods such as AND/OR search [2, 3] and integer linear programming (ILP) [1]guarantee optimality but are prohibitively expensive for large instances. Approximate methods offer scalability but often compromise on solution quality or consistency, especially in domains that demand high-precision inference.

Conditioning on a variable subset—fixing variables to specific values—is a common strategy for improving the tractability of inference by simplifying the underlying problem structure. This reduction in complexity can significantly shrink the search space and improve solver performance. This principle underlies classical techniques like cutset conditioning [4] and recursive conditioning [5], as well as strong branching heuristics [6, 7] in ILP solvers. A related concept in statistical physics is *decimation* [8–12], which iteratively fixes high-impact variables to reduce the search space in message-passing algorithms such as Survey Propagation, progressively simplifying the problem instance. However, the

effectiveness of these methods is highly sensitive to the choice and ordering of conditioned variables; poor decisions can lead to significant degradation in solution quality.

This leads to a fundamental question: *which variables should be fixed, to which values, and in what order, to simplify inference while preserving the optimal solution?* Fixing variables incorrectly can irrevocably exclude the true optimal solution, whereas deferring all conditioning preserves completeness but forfeits the computational benefits of simplification. Ideally, conditioning decisions should be both *safe*−i.e., aligned with optimal solutions and *useful*−i.e., significantly reduce computational effort.

In principle, access to the complete set of MPE solutions for a given instance would enable the identification of such beneficial assignments. For instance, if a variable assumes the same value across all optimal solutions, fixing it would preserve optimality and likely simplify the subsequent problem. In practice, however, enumerating all MPE solutions is infeasible [13], and exhaustively evaluating the impact of every variable-value pair on solver performance is prohibitively expensive. Consequently, existing methods rely on handcrafted heuristics [14, 15] or structure-based metrics [16] that often lack generalization.

In this work, we introduce *learning to condition* (L2C), a *data-driven* approach to rank variable-value assignments based on their utility for conditioning. The goal is to train a neural network to identify assignments that strike a balance between two objectives: preserving access to optimal solutions and simplifying the inference process. To achieve this, we train an attention-based neural network that assigns two scores to each variable-value assignment: an *optimality score*, estimating the likelihood of the assignment's presence in an optimal solution, and a *simplification score*, measuring the extent to which fixing the assignment reduces solver effort.

We obtain supervision for these scores via a scalable data generation pipeline. For each training instance, we solve the MPE query once using an oracle and treat the resulting solution as a proxy for the set of optimal solutions. Variable-value pairs present in the solution are labeled as positive examples for the optimality head. To supervise the simplification head, we fix individual variables to their optimal values, re-solve the query, and record solver statistics such as runtime and the number of explored nodes, providing a quantitative measure of the reduction in inference complexity.

A key insight of our method is that solution ambiguity modulates conditioning risk. If a variable takes the same value across all optimal solutions, fixing it is highly effective but also risky if the estimate is incorrect, as it could eliminate the true MPE solution. Conversely, when a variable's values are more evenly distributed, conditioning poses less risk. Even if the model slightly misestimates the correct value, the solution space is more tolerant, and optimality may still be preserved. Our model implicitly learns to navigate this trade-off by generating soft scores that reflect both expected utility and uncertainty, shaped through diverse training examples.

At inference time, the model assigns scores to all variable-value pairs in the query set for the given instance and conditions on those with the highest scores. This strategy incrementally simplifies the problem while avoiding premature elimination of the optimal solution. Additionally, the learned model can also be integrated into branch-and-bound solvers as node and variable selection heuristics.

**Contributions.** This paper introduces a neural network-based conditioning strategy for MPE inference in PGMs, with the following key contributions:

- We formalize *learning to condition* (L2C) as a scoring problem over variable-value pairs that jointly optimizes for solution preservation and inference tractability.
- We design a data-efficient supervision strategy that generates labels using oracle solutions and solver statistics, avoiding the need for exhaustive MPE enumeration.
- We introduce an attention-based architecture that generalizes across instances and yields informative optimality and simplification scores.
- We demonstrate that our method improves both inference efficiency and solution quality over classical heuristics across a range of benchmark PGMs.

Our results show that L2C enables scalable, learned conditioning decisions that adapt to instance structure, outperforming traditional heuristics in both speed and accuracy. Our implementation, solver integrations, and experiment scripts are publicly available at, `https://github.com/brijml/L2C`.

## 2 Background

We assume, w.l.o.g., that all random variables are binary and take values in $\{0, 1\}$. Individual variables are denoted by uppercase letters (e.g., $X$), and their assignments by corresponding lowercase letters (e.g., $x$). Sets of variables are denoted by bold uppercase letters (e.g., $\mathbf{X}$), and assignments to those sets are written in bold lowercase (e.g., $\mathbf{x}$). Given a full assignment $\mathbf{x}$ to a set $\mathbf{X}$, and a subset $\mathbf{Y} \subseteq \mathbf{X}$, we use $\mathbf{x_Y}$ to denote the projection of $\mathbf{x}$ onto $\mathbf{Y}$.

A **probabilistic graphical model** (PGM) $\mathcal{M} = \langle \mathbf{X}, \mathbf{F}, G \rangle$ is defined over a set of variables $\mathbf{X}$, a collection of log-potentials $\mathbf{F} = \{f_1, \ldots, f_m\}$, and an undirected primal graph $G = (\mathcal{V}, \mathcal{E})$. Each log-potential $f_i \in \mathbf{F}$ is defined over a subset $\mathbf{S}(f_i) \subseteq \mathbf{X}$, known as its scope. The vertex set $\mathcal{V}$ contains one node for each variable in $\mathbf{X}$, and an edge $(\mathcal{V}_a, \mathcal{V}_b) \in \mathcal{E}$ is added whenever $X_a$ and $X_b$ co-occur in the scope of some $f_i$. The model defines the following joint probability distribution:

$$P_{\mathcal{M}}(\mathbf{x}) \propto \exp\left( \sum_{f \in \mathbf{F}} f(\mathbf{x}_{\mathbf{S}(f)}) \right)$$

We focus on the **most probable explanation (MPE)** task: given observed evidence variables $\mathbf{E} \subset \mathbf{X}$ with assignment $\mathbf{e}$, find the most likely assignment for query variables $\mathbf{Q} = \mathbf{X} \setminus \mathbf{E}$. Formally, this is:

$$\mathrm{MPE}(\mathbf{Q}, \mathbf{e}) = \arg\max_{\mathbf{q}} P_{\mathcal{M}}(\mathbf{q} \mid \mathbf{e}) = \arg\max_{\mathbf{q}} \sum_{f \in \mathbf{F}} f((\mathbf{q}, \mathbf{e})_{\mathbf{S}(f)})$$

This problem is NP-hard in general and remains intractable for numerous model families [17–21].

Exact MPE algorithms include bucket elimination [22], which uses variable elimination via local reparameterization, and Mixed Integer Programming (MIP) encodings solved with branch-and-bound solvers like SCIP [23] that combine LP relaxations with systematic search. Such solvers are often anytime, providing progressively better solutions and bounds, with optimality certificates upon convergence. Nevertheless, for many real-world PGMs—particularly those with high treewidth—exact inference remains computationally infeasible.

**Conditioning** is a classical strategy for simplifying inference by fixing a subset of variables. For MPE, assigning values $\mathbf{q}_d$ to a conditioning set $\mathbf{Q}_d \subseteq \mathbf{Q}$ reduces the problem to solving MPE for the remaining variables $\mathbf{Q}_0 = \mathbf{Q} \setminus \mathbf{Q}_d$ given evidence $\mathbf{e} \cup \mathbf{q}_d$. The resulting residual solution $\mathbf{q}_0^*$ and $\mathbf{q}_d$ form the complete MPE solution. Conditioning is fundamental to exact methods like cutset conditioning [4], recursive conditioning [5], and search algorithms such as Branch and Bound (B&B) and AND/OR Branch and Bound (AOBB) [24]. Conditioning also informs heuristic methods such as decimation strategies inspired by statistical physics [9, 10, 25, 26].

**Branch and Bound (B&B)** performs a depth-first or best-first search over a decision tree, recursively partitioning the assignment space and pruning subproblems using lower bounds from LP relaxations or mini-bucket elimination [27]; its efficiency hinges on branching heuristics and bound quality. **AND/OR Branch and Bound (AOBB)** [24] improves upon B&B by exploiting graphical model structure through a *pseudo tree* and an AND/OR search graph. This graph uses OR nodes for variable choices and AND nodes for decomposition points, enabling independent search on conditionally independent components formed by assignments. AOBB further avoids redundant computations by merging nodes with identical separator contexts. Consequently, AOBB can prune substantial portions of the search space, often achieving exponential savings over standard B&B.

Graph-based methods like Mini-Bucket Elimination [27] generate bounds (e.g., upper bounds for MPE) that accelerate B&B and AOBB through earlier pruning. Variable and value ordering heuristics further enhance their effectiveness [28–30].

Recent efforts integrate learning into solver decisions. In MIP, *full strong branching* [7, 6] uses one-step look-ahead for selecting branch variables that maximize bound improvement; neural networks trained via imitation learning can replicate this [31, 32]. Other machine learning approaches include using SVMs to identify optimal decision variables [33] and NNs for effective node-selection strategies [34]. Recent neural approaches to MPE inference have concentrated on optimizing relaxed likelihood objectives [35–37]. While these methods eliminate the need for supervision, they provide no guarantees of optimality, as they rely on continuous relaxations of the MPE objective.

Our work advances *conditioning-based simplification* by learning a scoring function that ranks variable-value assignments according to their utility in reducing inference complexity while preserving optimality.

## 3  Learning to Condition (L2C)

We propose a neural approach for ranking variable-value assignments based on their utility in simplifying MPE inference while preserving optimality. This task, which we call **Learning to Condition (L2C)**, balances two goals: minimizing inference cost and avoiding exclusion of optimal solutions. Our model learns two scores per assignment: one measuring consistency with optimal solutions and another estimating simplification benefit. These data-driven scores guide the selection of high-confidence assignments and can also be used to steer node selection in B&B solvers. Next, we describe the L2C pipeline, including data generation, neural architecture, training objectives, ranking mechanism, and integration with inference procedures.

### 3.1  Data collection

We construct a dataset of MPE queries and solver outcomes to train the L2C model to score variable-value assignments based on two criteria: their consistency with optimal solutions and their ability to simplify inference when fixed. Because enumerating all optimal solutions or exhaustively evaluating all assignments is computationally infeasible, we collect targeted supervision using selective oracle queries under randomized input conditions.

The full data collection procedure is described in Algorithm 1. Given a probabilistic graphical model $\mathcal{M} = \langle \mathbf{X}, \mathbf{F}, G \rangle$, we repeatedly sample full assignments $\mathbf{x} \sim P_{\mathcal{M}}$ and randomly partition the variable set into disjoint subsets: a query set $\mathbf{Q} \subset \mathbf{X}$ of size $qr \cdot |\mathbf{X}|$, and an evidence set $\mathbf{E} = \mathbf{X} \setminus \mathbf{Q}$. The evidence assignment $\mathbf{e}$ is formed by projecting $\mathbf{x}$ onto $\mathbf{E}$, and an MPE solver is invoked to compute the optimal solution $\mathbf{q}^* = \arg\max_{\mathbf{q}} P_{\mathcal{M}}(\mathbf{q} \mid \mathbf{e})$. The resulting assignment, along with runtime and number of nodes explored, is stored as supervision.

To evaluate the impact of variable-value assignments on inference, we randomly sample a subset $\mathbf{C} \subset \mathbf{Q}$ of size at most $c_{\max}$, and for each $C \in \mathbf{C}$, consider both binary assignments $C = 0$ and $C = 1$. For each assignment, we augment the evidence to form $\mathbf{e}' = \mathbf{e} \cup \{C = c\}$, solve the resulting MPE query, and record solver statistics such as runtime, number of nodes explored, and objective value. These evaluations provide supervision for both the simplification and optimality scoring heads.

Evaluating all possible variable-value assignments for each instance is computationally prohibitive, especially for large models. To manage this cost, we introduce a sampling parameter $c_{\max}$ that bounds the number of variables considered for conditioning per instance. This allows us to gather informative supervision while keeping the number of additional MPE queries tractable. In practice, $c_{\max}$ can be chosen based on the model size, the characteristics of the solver, and the available compute budget.

When the solver fails to return a result within the allocated time budget $B$, we log surrogate statistics such as LP-bound improvements or relaxed objective scores. Although these surrogates may be noisy, they increase coverage across instances and improve the generalization ability.

---

**Algorithm 1:** Data Collection for L2C with Conditioning Assignments

---

**Input:** PGM $\mathcal{M} = \langle \mathbf{X}, \mathbf{F}, G \rangle$, Query Ratio $qr$, Max Conditions $c_{\max}$, Budget $B$
**Output:** Dataset $DB$
**Function** SolveMPE($\mathcal{M}, \mathbf{e}, B$):
    Run MPE solver on $\mathcal{M}$ with evidence $\mathbf{e}$ and time bound $B$
    **return** rec $= \{\text{assign}, \text{time}, \#\text{nodes}\}$

Initialize $DB \leftarrow \emptyset$
**while** *not enough samples in $DB$* **do**
    Sample $\mathbf{x} \sim P_{\mathcal{M}}$
    $\mathbf{Q} \leftarrow$ random subset of $\mathbf{X}$ of size $qr|\mathbf{X}|$
    $\mathbf{E} \leftarrow \mathbf{X} \setminus \mathbf{Q}$
    $\mathbf{e} \leftarrow \mathbf{x}_{\mathbf{E}}$
    rec $\leftarrow$ SolveMPE($\mathcal{M}, \mathbf{e}$)
    Store $(\mathbf{e}, \text{rec})$ in $DB$
    $\mathbf{C} \leftarrow$ random subset of $\mathbf{Q}$ of size $c_{\max}$
    **for** *each $C \in \mathbf{C}$* **do**
        **for** *each value $c \in \{0, 1\}$* **do**
            $\mathbf{e}' \leftarrow \mathbf{e} \cup \{C = c\}$
            rec$' \leftarrow$ SolveMPE($\mathcal{M}, \mathbf{e}'$)
            Store $(\mathbf{e}', \text{rec}')$ in $DB$

**return** $DB$

---

Each $\mathbf{e}$ and its corresponding optimal assignment $\mathbf{q}^*$ (if returned by the MPE solver) in $DB$ serves as supervision for the optimality head. For simplification, we convert the collected solver statistics into a ranking distribution over a small set of candidate variables $\mathbf{C} \subset \mathbf{Q}$. For each candidate $C \in \mathbf{C}$, we

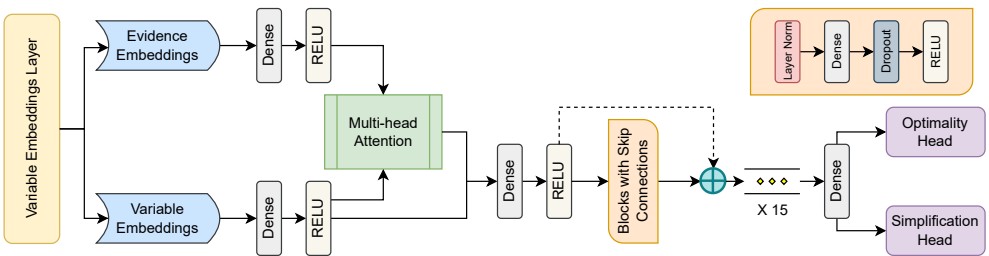

Figure 1: Attention-based architecture for scoring variable-value pairs by their utility in simplifying MPE inference while preserving optimality.

define: $p_C \propto \frac{1}{t_C}$ where $t_C$ is a composite scalar derived from multiple solver statistics recorded for the query conditioned on $C = c$, including runtime, number of search nodes explored, and objective value. These quantities are combined linearly to reflect the overall simplification utility of the assignment. The resulting distribution is normalized using a softmax over inverse scores, assigning higher probabilities to variable-value assignments that are more effective at reducing inference cost.

Thus the resulting dataset contains both exact and surrogate supervision, allowing the model to learn to balance optimality preservation with inference simplification across a range of graphical model instances.

### 3.2  Neural network architecture

For a partially observed MPE instance (evidence $\mathbf{E} \subset \mathbf{X}$ with assignment $\mathbf{e}$, query variables $\mathbf{Q} = \mathbf{X} \setminus \mathbf{E}$), our model (Figure 1) scores variable-value pairs on their potential for MPE optimality preservation and inference cost reduction. Each pair $(X_i = x_i)$ is represented by an embedding vector [38]. For binary variables, an embedding table of size $2 \times |\mathbf{X}|$ provides distinct embeddings for each variable's possible values. An additional embedding indicates variable status (observed/evidence or unobserved/query), enabling support for arbitrary evidence-query partitions.

For each unobserved variable $Q_i \in \mathbf{Q}$, its two value assignment embeddings query a multi-head attention module [39] using evidence variable embeddings as keys and values, contextualizing each candidate assignment relative to the instance and enabling the model to learn associations between evidence and promising assignments. The output contextualized embeddings, concatenated with their original counterparts, pass through a shared encoder (comprising fully connected layers with ReLU activations, residual connections [40], and dropout [41]). This encoder produces a representation for each variable-value pair, which is then passed to two separate prediction heads.

The *optimality head*, a two-layer MLP with a sigmoid output, estimates an assignment's likelihood of inclusion in the MPE solution. Trained with binary cross-entropy loss against oracle-derived labels, this head learns a relaxed optimality proxy, capturing generalizable patterns across problem instances.

A separate *simplification head*, also a two-layer MLP, outputs unnormalized scores for candidate assignments. These scores, after softmax normalization, are trained via a list-ranking cross-entropy loss [42], using targets derived from solver statistics (Section 3.1) like runtime, search depth, or relaxed objectives. By predicting relative utility instead of absolute complexity reduction, this head produces scores that generalize well and exhibit noise robustness.

Our architecture exhibits permutation invariance to variable ordering and supports arbitrary evidence-query partitions, promoting generalization. Cross-attention facilitates query-specific evidence conditioning, while the dual-head design jointly optimizes solution quality and inference efficiency. This yields a flexible scoring function adept at guiding conditioning during inference.

### 3.3  Loss functions

We train our model using a multi-task objective [43] that combines two losses: a binary classification loss for predicting whether a variable-value pair is part of an optimal MPE solution, and a list-ranking loss that encourages the model to prioritize assignments that simplify inference. These respectively supervise the network's optimality and simplification heads.

**Optimality loss.** For each query variable $Q_i \in \mathbf{Q}$ and its possible values $q_i \in \{0, 1\}$, the optimality head predicts $\hat{y}_i^{(q)} \in [0, 1]$, the probability that assignment $Q_i = q$ is in the oracle MPE solution, where $y_i^{(q)} \in \{0, 1\}$ is the true label. The binary cross-entropy loss is applied independently per pair:

$$\mathcal{L}_{\text{opt}} = -\sum_{Q_i \in \mathbf{Q}} \sum_{q \in \{0,1\}} \left( y_i^{(q)} \log \hat{y}_i^{(q)} + (1 - y_i^{(q)}) \log(1 - \hat{y}_i^{(q)}) \right)$$

This formulation supports supervision with multiple optimal assignments, enabling the model to learn from all optimal solutions for a given instance.

**Simplification ranking loss.** For each instance, the training dataset contains solver statistics for a subset of query variables $\mathbf{C} \subset \mathbf{Q}$, obtained during data collection as described in Section 3.1. For each $C \in \mathbf{C}$, we construct a target probability distribution $p_C$ that reflects the simplification utility of fixing $C = c$, based on a combination of solver statistics such as runtime, number of nodes explored, and objective value. The simplification head generates scores for all variable-value pairs in $\mathbf{Q}$, which are normalized via softmax into a predicted distribution $\hat{p}_C$. We then compute a list-ranking cross-entropy loss [42]:

$$\mathcal{L}_{\text{rank}} = -\sum_{C \in \mathbf{C}} p_C \log \hat{p}_C$$

This loss encourages the model to prioritize assignments that lead to greater reductions in inference cost. Since supervision is available only for a subset of $\mathbf{Q}$, we apply a binary mask to restrict the loss computation to the subset $\mathbf{C}$ for which solver statistics were recorded during training. This ensures that only observed variable-value pairs influence parameter updates, while predictions for the rest of $\mathbf{Q}$ remain unconstrained.

**Joint objective and masking.** The total loss combines these components as a weighted sum:

$$\mathcal{L} = \lambda_{\text{opt}} \cdot \mathcal{L}_{\text{opt}} + \lambda_{\text{rank}} \cdot \mathcal{L}_{\text{rank}}$$

Here, $\lambda_{\text{opt}}$ and $\lambda_{\text{rank}}$ balance the objectives' relative importance. We train using both exact and surrogate datasets. If supervision is available for only one head (optimality or ranking), we apply its loss and mask the other, preventing gradient updates from missing targets. This joint objective allows the model to balance accuracy with inference efficiency by leveraging both exact and approximate supervision.

### 3.4 Inference-time strategies

At inference time, the model guides variable-value assignment decisions that simplify the MPE problem before the solver is invoked. Given a probabilistic graphical model $\mathcal{M} = (\mathbf{X}, \mathcal{F})$, an initial evidence set $\mathbf{E} \subset \mathbf{X}$, and assignment $\mathbf{e}$, the model takes the current evidence as input and produces scores for each remaining query variable-value pair $(Q_i = q_i)$, where $Q_i \in \mathcal{V} \setminus \mathbf{E}$ and $q_i \in \{0, 1\}$.

---

**Algorithm 2:** Beam Search for L2C

**Input:** PGM $\mathcal{M}$, Scoring function $\mathcal{F}$, Initial evidence $\mathbf{E}$, Time limit $tl$, Max depth $\mathcal{D}_{\max}$, Beam width $W$

**Output:** Approximate MPE solution

Init. beam $B \leftarrow \{(0, \mathbf{E})\}$ //(score,evid)
**for** $d = 1$ *to* $\mathcal{D}_{\max}$ **do**
    Initialize candidate list $C \leftarrow \emptyset$
    **foreach** $(s, \mathbf{E}_{partial}) \in B$ **do**
        Identify query variables
        $\mathbf{Q}_{\text{rem}} = \mathbf{X} \setminus \mathbf{E}_{\text{partial}}$
        **foreach** *unfixed* $Q_i \in \mathbf{Q}_{rem}$ **do**
            **foreach** *value* $q_i \in \{0, 1\}$ **do**
                Let
                $\mathbf{E}' \leftarrow \mathbf{E}_{\text{partial}} \cup \{Q_i = q_i\}$
                Let $s' \leftarrow \mathcal{F}(\mathbf{E}')$
                Add $(s', \mathbf{E}')$ to $C$

    Sort $C$ in descending order by score $s'$
    Prune $C$ to retain top $W$ elements
    $B \leftarrow C$
Let $(s^*, \mathbf{E}^*) \leftarrow \text{TopCandidate}(B)$
Run $\mathbf{q}^* \leftarrow \text{SolveMPE}(\mathcal{M}, \mathbf{E}^*, tl)$
**return** $\mathbf{q}^*$

---

The *optimality head* outputs confidence $\hat{y}_i^{(q)} \in [0, 1]$ for an assignment's MPE consistency; the *simplification head* provides a score $s_i^{(q)} \in \mathbb{R}$ for its inference simplification utility. These scores are combined to select conditioning assignments via two strategies:

**Greedy conditioning.** This strategy iteratively filters assignments by an optimality threshold $\tau$, selects the one with the highest simplification score among confident candidates, and adds it to evidence. This repeats for a fixed number of steps or until no confident options remain, before passing the simplified MPE query to a solver.

**Beam search.** To explore multiple conditioning sequences, beam search (Algorithm 2) maintains $W$ partial assignment sequences. At each step $d \leq \mathcal{D}_{\max}$, sequences are expanded with assignments scored as in greedy conditioning. The best final sequence provides evidence $\mathbf{E}^*$ for the MPE solver.

**Final solution.** Let $\mathbf{E}^*$ denote the final evidence set selected by either strategy. The residual MPE query is defined over $\mathbf{Q}_{\text{residual}} = \mathcal{V} \setminus \mathbf{E}^*$. We call an MPE solver (exact or anytime) to compute:

$$\mathbf{q}^*_{\text{residual}} = \arg\max_{\mathbf{q}} P_{\mathcal{M}}(\mathbf{q} \mid \mathbf{e}^*)$$

and return the complete solution $\mathbf{q}^* = \mathbf{q}^*_{\text{residual}} \cup \mathbf{e}^*$.

This framework uses learned assignments to restructure the MPE problem, reducing solver complexity while preserving correctness and enabling time-bounded inference.

**NN-Guided branch-and-bound search** Our neural model also enhances branch-and-bound (B&B) solvers. It guides branching by selecting variables predicted to simplify the subproblem (using scores akin to greedy conditioning) and uses optimality head predictions to tighten lower bounds. These actions accelerate B&B convergence while maintaining optimality guarantees.

Learning to Condition (L2C) transforms the traditionally static variable selection problem into a learned, instance-specific decision process. Compatible with both exact and approximate solvers, L2C generalizes across problem instances and enables principled trade-offs between solution quality and computational cost. We next empirically evaluate its performance across a diverse set of benchmarks and graphical models.

## 4 Experiments

In this section, we evaluate our neural strategies, L2C-OPT (using only the optimality head for scoring) and L2C-RANK (using the scoring procedure from Section 3.4). We benchmark these strategies for two tasks: (1) against standard conditioning heuristics, and (2) as branching and node selection methods within B&B solvers. We begin by describing our experimental framework, including the PGMs, baseline methods, evaluation metrics, and NN architecture.

### 4.1 Graphical models

We evaluated our method and baselines on 14 high-treewidth binary probabilistic graphical models (PGMs) from UAI inference competitions [44, 45], featuring 90 to 1444 variables and up to 1444 factors. Using Gibbs sampling [1], we generated 12,000 training, 1,000 test, and 1,000 validation examples per model. All MPE instances used 75% of the PGM's variables as query variables. We construct the supervised dataset following the procedure outlined in Section 3.1. During data generation, QR is set to 0.75 and $c_{\max}$ is set to 10% of the total number of variables in each PGM.

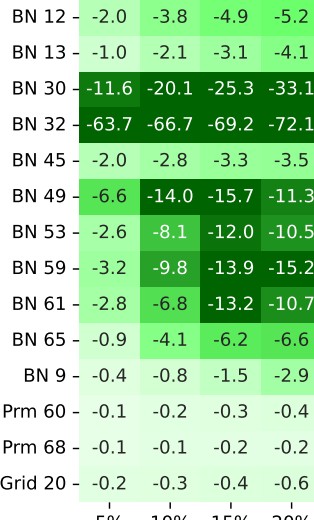

(a) Wins vs. oracle by time limit for all methods (G: Graph, SB: Full Strong Branching, LO: L2C-OPT, LR: L2C-RANK).

(b) L2C-RANK vs. oracle $\mathcal{LL}$ gap by $\mathcal{D}_{max}$ (cols) and graphical model (rows).

Figure 2: Neural vs. baseline methods for greedy conditioning (SCIP oracle). (a) Win counts; (b) Log-likelihood gaps. Color: Darker green = stronger, darker red = weaker performance. Grey: Timeouts (30s).

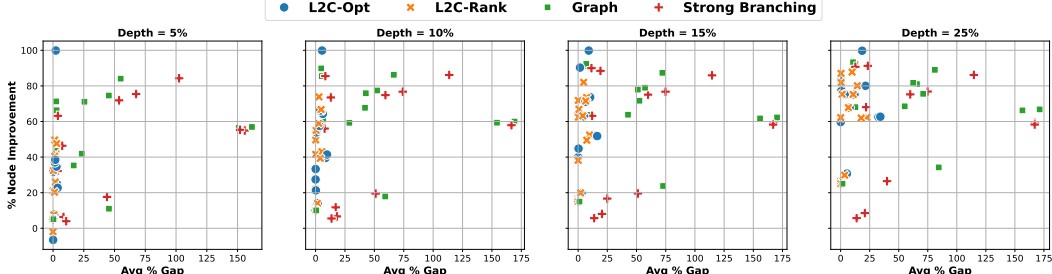

Figure 3: Greedy conditioning methods (AOBB oracle): Average solution gap (x-axis) versus % node reduction (y-axis). Each subfigure denotes a fixed decision count.

## 4.2 Experimental setup and methods

The neural networks in L2C-OPT and L2C-RANK use 256-dimensional embeddings, two multi-head attention layers, and 15 skip-connection blocks. Dense layers have 512 units with 0.1 dropout [41] and ReLU activations. We trained the models using Adam [46] with a learning rate of $8 \times 10^{-4}$, an exponential decay rate of 0.97, a batch size of 128, and early stopping after 5 stagnant epochs (maximum 50 epochs). All models were implemented in PyTorch [47] and executed on an NVIDIA A40 GPU. Appendix B provides further hyper-parameter details.

**Baseline methods:** We compare our approach with two heuristic methods: The **graph-based heuristic** [28–30] prioritizes variables by degree, selecting them in descending order after removing evidence nodes. This max-degree strategy assigns each variable its most likely value, estimated via Gibbs Sampling [1, 48] over the PGM. The **full strong branching heuristic** [7], a classical technique for enhancing branch-and-bound efficiency, formulates the MPE query as an ILP [1]. It then scores variable-value pairs by their impact on search space pruning.

**Oracle:** We use two oracles on unconditioned queries and as final solvers post-conditioning: the SCIP solver [23], and the AND/OR Branch and Bound (AOBB) solver (per Otten and Dechter [24] and implemented by Otten [3]). MPE queries are encoded as Integer Linear Programs (ILPs) [1] for SCIP, which applies its branch-and-bound algorithm subject to a 60-second time limit per query.

**Evaluation criteria:** The competing approaches were evaluated based on two criteria: log-likelihood scores and runtime. The log-likelihood scores, calculated as $\ln p_{\mathcal{M}}(\mathbf{e}, \mathbf{q})$, assesses the quality of the solution in terms of optimality. The runtime measures the efficiency of the conditioning process by indicating how quickly each method solves the problem.

## 4.3 Empirical evaluations

### 4.3.1 Greedy conditioning

We evaluate our methods and baselines as scoring functions ($\mathcal{F}$) in a greedy conditioning framework (Section 3.4), selecting up to 25% of query variables.[1] For each PGM, we test 12 configurations: four conditioning depths (5%, 10%, 15%, 25% of query variables) combined with three oracle time budgets (10s, 30s, 60s). Post-conditioning, SCIP attempts to solve the residual problem within the allotted budget.

Figure 2a (heatmap) quantifies for each conditioning strategy the number of the 12 configurations where it enabled the SCIP oracle to achieve a higher average log-likelihood ($\mathcal{LL}$) score than solving the original, non-conditioned query. The oracle's time budget was identical for both conditioned and non-conditioned queries.

The results (Figure 2a) show that both L2C-OPT (LO) and L2C-RANK (LR) consistently enhance oracle performance over direct solving of the original query. L2C-RANK performs best, frequently

---

[1]To ensure a fair comparison, as the max-degree baseline supports only a beam size of 1, we use greedy conditioning for neural and strong branching methods. Appendix E provides beam search results.

surpassing the non-conditioned oracle across all 12 configurations. Conversely, full strong branching (SB) and the graph-based heuristic (G) only occasionally improve $\mathcal{LL}$ scores.

Figure 2b presents the average percentage $\mathcal{LL}$ gap: $\frac{1}{N} \sum_{i=1}^{N} (\mathcal{LL}_S^{(i)} - \mathcal{LL}_D^{(i)})/(|\mathcal{LL}_S^{(i)}|) \times 100$, where $\mathcal{LL}_S^{(i)}$ and $\mathcal{LL}_D^{(i)}$ are the oracle's log-likelihoods for instance $i$ without and with conditioning by L2C-RANK, respectively. Rows denote PGMs, and columns indicate conditioning depth (percentage of query variables conditioned). The oracle's time budget is fixed at 30s for both approaches.

Predominantly negative (green) values in Figure 2b indicate that L2C-RANK conditioning helps the oracle find superior solutions within the 30s budget. The gap becomes increasingly negative with more conditioning decisions (i.e., greater conditioning depth), demonstrating our method's scalability. This improvement occurs because conditioning simplifies the query by instantiating additional variables, thus presenting an easier query to the oracle.

**Assisting AOBB via conditioning:** We also evaluated our neural heuristic with the AND/OR Branch and Bound (AOBB) solver [3] as an alternative oracle.

Figure 3 plots the percentage reduction in AOBB's processed node count (y-axis, proxy for effort reduction) against the average $\mathcal{LL}$ gap from the exact MPE solution (x-axis).[2] Points near the y-axis indicate high solution quality with reduced effort; greater x-values imply more quality degradation.

Our conditioning strategies, as shown in Figure 3, yield near-optimal solutions while substantially reducing AOBB's processed node count. In contrast, traditional baselines exhibit larger solution gaps with increased conditioning depth due to increasingly suboptimal decisions. Our methods consistently maintain high solution quality (most points near the y-axis) and more effectively accelerate AOBB by shrinking the search space, particularly with deeper conditioning.

Consequently, L2C enables efficient resolution of queries that are otherwise challenging for the oracle, especially for large-treewidth models, frequently yielding substantially better solutions under identical time constraints.

Figure 4: Heuristic comparison across datasets (rows) and time limits (columns); darker green/red indicates better/worse L2C-RANK performance.

### 4.3.2 NN-Guided branch-and-bound search

We also employ our neural network outputs to guide branch-and-bound search, using them as branching rules [32] and node selection heuristics [34], following the methodology detailed in Section 3.4.

Figure 4 compares our neural-guided heuristics against SCIPs default strategies [23, 49–51], showing the average $\mathcal{LL}$ gap (defined previously) over three time limits and all datasets. The predominantly green cells, particularly dark green ones, in the heatmap signify that our neural heuristics often outperform SCIP's defaults. Furthermore, the few red cells underscore that our approach generally matches or surpasses SCIP's default heuristic performance and delivers better solutions within identical time budgets.

### 4.3.3 Comparing conditioning methods by per-decision time

Table 1 reports the average time required to make a single conditioning decision across all networks. Both L2C-OPT and L2C-RANK maintain consistent decision times across different network sizes, demonstrating the scalability and practical feasibility of our learned heuristics. In contrast, the graph-based heuristic exhibits increasing latency as network size grows, while strong branching is considerably slower and often infeasible for large models. We imposed a 30-second timeout per decision, under which strong branching failed to return results for the largest networks (BN 30 and

---

[2]Node and $\mathcal{LL}$ gap reductions for AOBB are relative to its unconditioned execution. AOBB's internal preprocessing and heuristics often enable it to solve MPE queries within the time limit; thus, a reduced node count with a near-zero $\mathcal{LL}$ gap signifies simplified inference without quality loss.

Table 1: Mean $\pm$ standard deviation of decision time (s) across networks for each branching strategy. Gray cells indicate cases where the 30 s timeout was reached.

| Network | L2C-OPT | L2C-RANK | Graph | Strong Branching |
|---|---|---|---|---|
| **BN 12** | $0.001 \pm 0.000$ | $0.002 \pm 0.001$ | $0.001 \pm 0.001$ | $2.215 \pm 0.161$ |
| **BN 9** | $0.003 \pm 0.002$ | $0.006 \pm 0.004$ | $0.007 \pm 0.003$ | $1.256 \pm 0.115$ |
| **BN 13** | $0.001 \pm 0.001$ | $0.002 \pm 0.001$ | $0.022 \pm 0.020$ | $2.787 \pm 0.114$ |
| **Grid 20** | $0.006 \pm 0.004$ | $0.011 \pm 0.004$ | $0.107 \pm 0.062$ | $7.325 \pm 0.117$ |
| **BN 65** | $0.005 \pm 0.005$ | $0.008 \pm 0.005$ | $0.086 \pm 0.053$ | $5.871 \pm 0.320$ |
| **BN 59** | $0.005 \pm 0.004$ | $0.009 \pm 0.005$ | $0.097 \pm 0.057$ | $10.993 \pm 1.683$ |
| **BN 49** | $0.007 \pm 0.005$ | $0.011 \pm 0.004$ | $0.100 \pm 0.059$ | $19.713 \pm 3.417$ |
| **BN 53** | $0.005 \pm 0.004$ | $0.009 \pm 0.005$ | $0.095 \pm 0.056$ | $9.106 \pm 0.993$ |
| **BN 61** | $0.009 \pm 0.004$ | $0.013 \pm 0.004$ | $0.103 \pm 0.063$ | $20.896 \pm 3.038$ |
| **BN 45** | $0.011 \pm 0.005$ | $0.014 \pm 0.003$ | $0.088 \pm 0.050$ | $8.675 \pm 0.733$ |
| **Prm 68** | $0.013 \pm 0.004$ | $0.016 \pm 0.003$ | $0.085 \pm 0.051$ | $4.279 \pm 0.118$ |
| **Prm 60** | $0.015 \pm 0.002$ | $0.015 \pm 0.002$ | $0.089 \pm 0.052$ | $5.432 \pm 0.270$ |
| **BN 30** | $0.018 \pm 0.002$ | $0.018 \pm 0.002$ | $0.098 \pm 0.058$ | — |
| **BN 32** | $0.023 \pm 0.005$ | $0.025 \pm 0.009$ | $0.102 \pm 0.060$ | — |

BN 32). Overall, these results show that L2C achieves low and predictable decision costs even as model complexity increases.

**Summary:** Our comprehensive experiments validate the consistent superiority of L2C. We evaluate our approach using two competitive and complementary solver backends: AOBB, a top-performing AND/OR search solver, and SCIP, a state-of-the-art ILP optimizer. This combination encompasses structurally distinct inference paradigms, highlighting the solver-agnostic advantages of learned conditioning. As a conditioning strategy, it empowers SCIP to achieve better solutions within time limits, especially for intractable problems, and allows AOBB to preserve solution quality while drastically reducing search effort (node exploration). Furthermore, employing our neural scores as branching and node selection heuristics enhances SCIP's solution quality under the same time constraints. Importantly, L2C maintains stable decision times across increasing network sizes, underscoring its scalability and suitability for large-scale inference. Collectively, these results affirm our method's effectiveness both as a conditioning strategy, improving solution quality and efficiency and as an effective heuristic guiding branch-and-bound solvers.

## 5    Conclusion

We introduced *Learning to Condition* (L2C), a neural approach leveraging solver search traces to accelerate Most Probable Explanation (MPE) inference in Probabilistic Graphical Models (PGMs). L2C learns to identify variable assignments that effectively balance inference simplification with optimality preservation, superseding handcrafted heuristics. It serves either as a pre-processor simplifying problems for exact solvers or as an internal policy guiding branching and variable selection within search algorithms. Evaluations on high-treewidth models show L2C substantially curtails the search space, achieving solution quality comparable or superior to baselines and thus providing a more efficient path to MPE solutions.

**Limitations and future work:** The current limitations lie in scalability, solver signal diversity, and generalization to unseen model structures. The method is tailored to a fixed PGM and depends on solver-derived supervision, which is costly to obtain at scale. Extending L2C to other solver families, such as MAXSAT, local search, or soft arc-consistency solvers, would necessitate substantial modifications to the architecture, data collection pipeline, and learning objectives.

Future work will focus on scaling L2C to handle larger models with millions of variables and factors, while employing more efficient and bootstrapped data collection techniques to reduce oracle dependence. We will also integrate richer solver signals (e.g., branch-and-cut statistics) to enhance pruning and design neural bounding mechanisms that extend learned guidance beyond conditioning. Finally, learned variable orderings will be explored for novel neural bounding strategies, extending guidance beyond conditioning to further boost inference performance.

## Acknowledgments and Disclosure of Funding

This work was supported in part by the DARPA CODORD program under contract number HR00112590089, the DARPA Assured Neuro Symbolic Learning and Reasoning (ANSR) Program under contract number HR001122S0039, the National Science Foundation grant IIS-1652835, and the AFOSR award FA9550-23-1-0239.

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

# A  Description of the probabilistic graphical models

Table 2 summarizes the networks used in our experiments, listing the number of variables and factors in each. The experiments were conducted on high-treewidth probabilistic graphical models (PGMs) of varying sizes, with the number of variables ranging from 90 to 1440 and factor counts covering a similar range. The table also lists the network aliases used throughout the paper for ease of reference.

Table 2: Summary of networks used in our experiments, including the number of variables and factors for each.

| Network Name | Network Alias | Number of Factors | Number of Variables |
|:---:|:---:|:---:|:---:|
| BN_12 | BN 12 | 90 | 90 |
| BN_9 | BN 9 | 105 | 105 |
| BN_13 | BN 13 | 125 | 125 |
| grid20x20.f5.wrap | Grid 20 | 1200 | 400 |
| BN_65 | BN 65 | 440 | 440 |
| BN_59 | BN 59 | 540 | 540 |
| BN_53 | BN 53 | 561 | 561 |
| BN_49 | BN 49 | 661 | 661 |
| BN_61 | BN 61 | 667 | 667 |
| BN_45 | BN 45 | 880 | 880 |
| Promedas_68 | Prm 68 | 1022 | 1022 |
| Promedas_60 | Prm 60 | 1076 | 1076 |
| BN_30 | BN 30 | 1156 | 1156 |
| BN_32 | BN 32 | 1440 | 1440 |

# B  Hyperparameter details

We used a consistent set of hyperparameters across all networks in our experiments. From the initial dataset of 14,000 samples, we allocated 13,000 for training and 1,000 for testing. The training set was further split into 12,000 samples for training and 1,000 for validation.

The neural networks in L2C-OPT and L2C-RANK use 256-dimensional embeddings, two multi-head attention layers, and 15 skip-connection blocks. Dense layers have 512 units with 0.1 dropout [41] and ReLU activations. We trained the models using Adam [46] with a learning rate of $8 \times 10^{-4}$, an exponential decay rate of 0.97, a batch size of 128, and early stopping after 5 stagnant epochs (maximum 50 epochs). All models were implemented in PyTorch [47] and executed on an NVIDIA A40 GPU. After training, we evaluated the model on 1,000 MPE queries with a query ratio of 0.75, defined as the fraction of variables in the query set.

Hyperparameters were selected via 5-fold cross-validation. In particular, the weighting parameters $\lambda_{opt}$ and $\lambda_{rank}$ were critical for balancing the preservation of optimal solutions with inference efficiency. We searched over $\lambda_{opt} \in \{0.3, 0.35, 0.4, 0.45, 0.5\}$ and set $\lambda_{rank} = 1 - \lambda_{opt}$.

# C  Extending L2C to multi-valued and continuous domains

While the main formulation of *Learning to Condition* (L2C) focuses on binary variables for simplicity, the framework generalizes naturally to multi-valued discrete and continuous domains with minor architectural and procedural modifications.

## C.1  Multi-valued discrete variables

For multi-valued variables, we define a variable-value pair for each possible assignment. During data collection, the same procedure is used to gather solver traces, now over the expanded set of candidate pairs. The attention-based scoring architecture remains unchangedeach (variable, value) pair is embedded independently and scored through the same optimality and simplification heads. This formulation allows L2C to reason jointly over variablevalue combinations without requiring major adjustments to training or inference. At inference time, greedy or beam-based selection can be

performed over all pairs, and conditioning proceeds as in the binary case. The learned heuristic thus generalizes directly to categorical domains.

## C.2 Continuous variables

Directly applying L2C to continuous domains is infeasible because the set of possible values is uncountable. To adapt, we discretize each continuous variable into a finite set of representative values (e.g., via uniform or quantile-based binning). The data collection process then conditions on these discretized values and records solver statistics as in the discrete case. During training, embeddings are created for each (variable, discretized value) pair, and the existing attention-based network and loss objectives are applied without modification. To mitigate discretization artifacts, soft-label targets or local smoothing can be used to ensure smooth transitions across adjacent bins.

At inference time, the model selects promising (variable, value) pairs based on the learned scores. After conditioning on discretized values, local continuous optimization methods (e.g., gradient descent, coordinate search) can optionally refine assignments within selected regions, improving precision beyond discretization resolution. This hybrid strategy allows L2C to extend its learned conditioning capabilities to continuous-variable inference while maintaining compatibility with existing solver frameworks.

## D   Greedy conditioning performance using SCIP as oracle

In this section, we analyze the conditioning performance of all methods under varying oracle time budgets and numbers of greedy conditioning decisions when SCIP solver [23] is used as the oracle. For each method and network, we report the average percentage gap in log-likelihood. The y-axis in Figures 5 to 18 shows the average percentage gap in log-likelihood scores, computed as:

$$\text{avg. \% gap} = \frac{1}{N} \sum_{i=1}^{N} \frac{(\mathcal{LL}_S^{(i)} - \mathcal{LL}_D^{(i)})}{|\mathcal{LL}_S^{(i)}|} \times 100 \tag{1}$$

where $\mathcal{LL}_S^{(i)}$ denotes the log-likelihood score of the oracle before conditioning, and $\mathcal{LL}_D^{(i)}$ denotes the score after conditioning for instance $i$. Thus, negative values indicate that conditioning improves solution quality, while positive values indicate a decline in performance. The x-axis represents the number of conditioning decisions, and each subfigure corresponds to a distinct oracle time budget.

As shown in the figures, our neural network-based strategies, L2C-OPT and L2C-RANK, consistently improve the oracle performance after conditioning, as evidenced by negative percentage gaps. In contrast, the **graph-based heuristic** and the **full strong branching heuristic** lead to improvements in only a small fraction of instances. Since these baselines do not produce optimal decisions, their performance generally worsens with increasing oracle time budgets, resulting in larger positive average gaps. This degradation becomes more pronounced as the number of conditioning decisions increases. In comparison, our approaches yield increasingly negative average gaps with more decisions, indicating consistent performance gains.

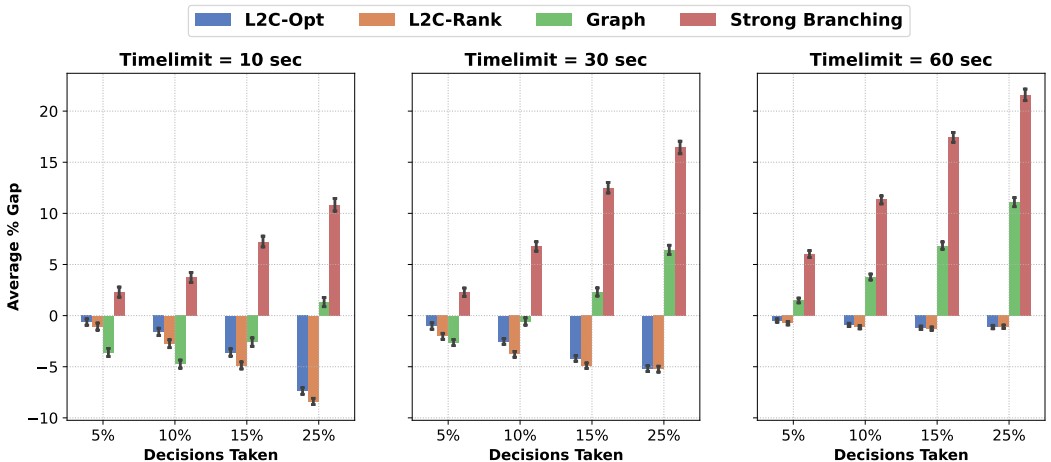

Figure 5: Average percentage gap on the BN 12 network for greedy conditioning using our methods, L2C-OPT and L2C-RANK, and baseline approaches across varying time budgets and numbers of decisions. More negative values indicate better performance.

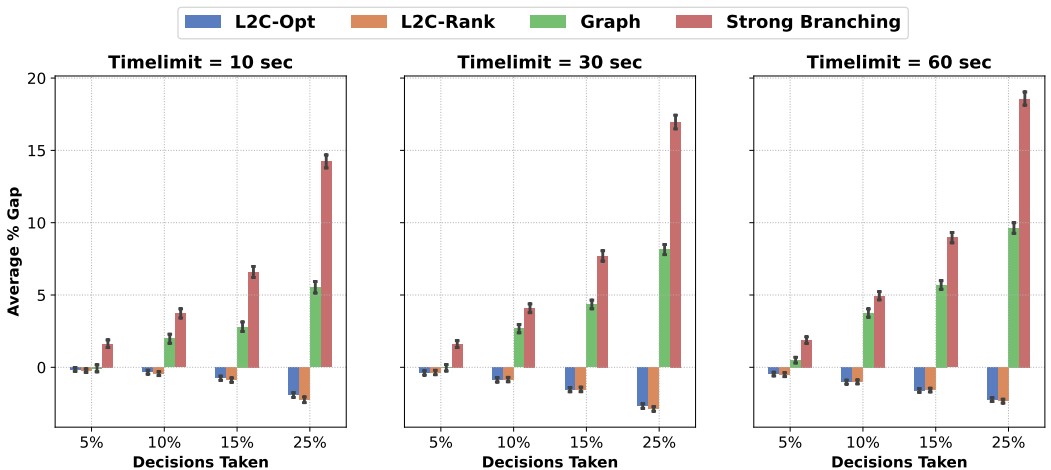

Figure 6: Average percentage gap on the BN 9 network for greedy conditioning using our methods, L2C-OPT and L2C-RANK, and baseline approaches across varying time budgets and numbers of decisions. More negative values indicate better performance.

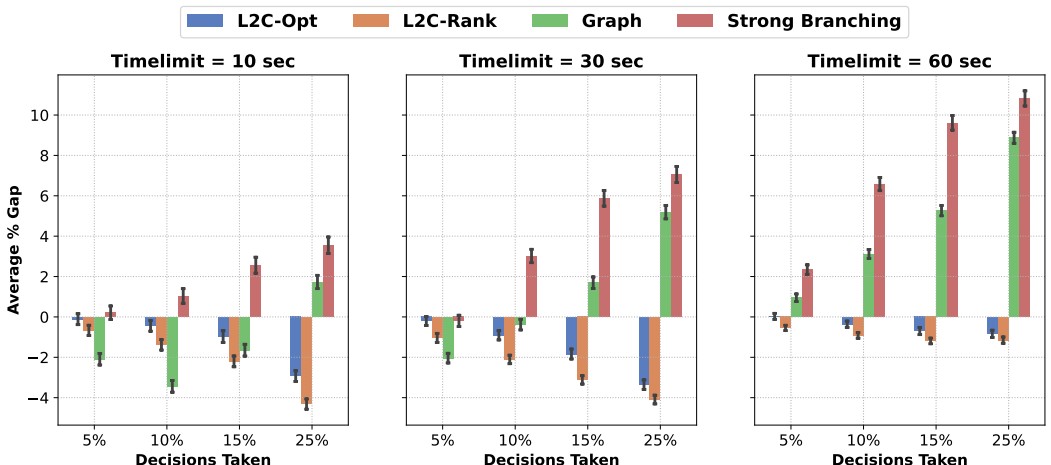

Figure 7: Average percentage gap on the BN 13 network for greedy conditioning using our methods, L2C-OPT and L2C-RANK, and baseline approaches across varying time budgets and numbers of decisions. More negative values indicate better performance.

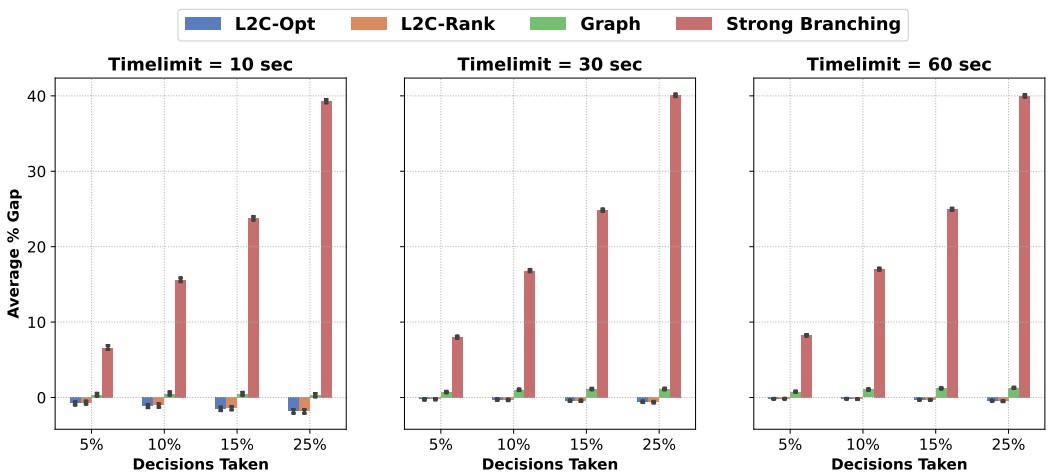

Figure 8: Average percentage gap on the Grid 20 network for greedy conditioning using our methods, L2C-OPT and L2C-RANK, and baseline approaches across varying time budgets and numbers of decisions. More negative values indicate better performance.

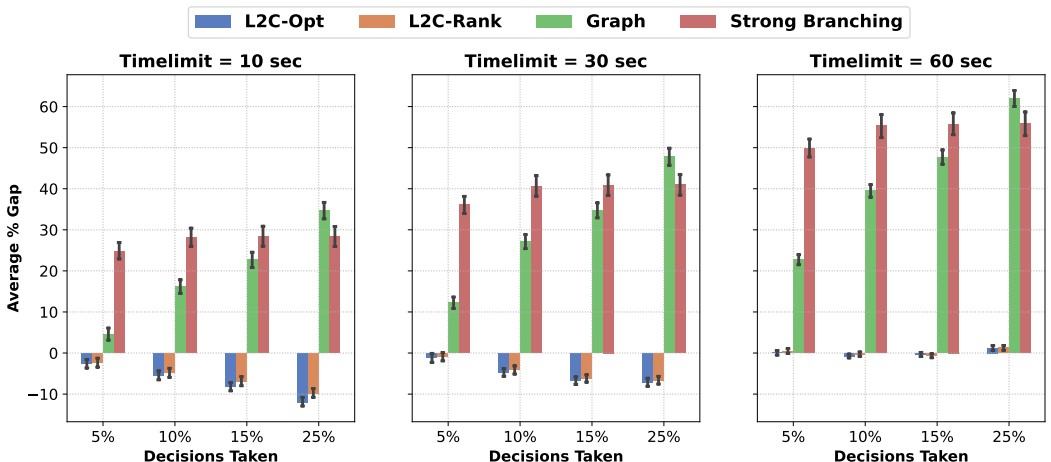

Figure 9: Average percentage gap on the BN 65 network for greedy conditioning using our methods, L2C-OPT and L2C-RANK, and baseline approaches across varying time budgets and numbers of decisions. More negative values indicate better performance.

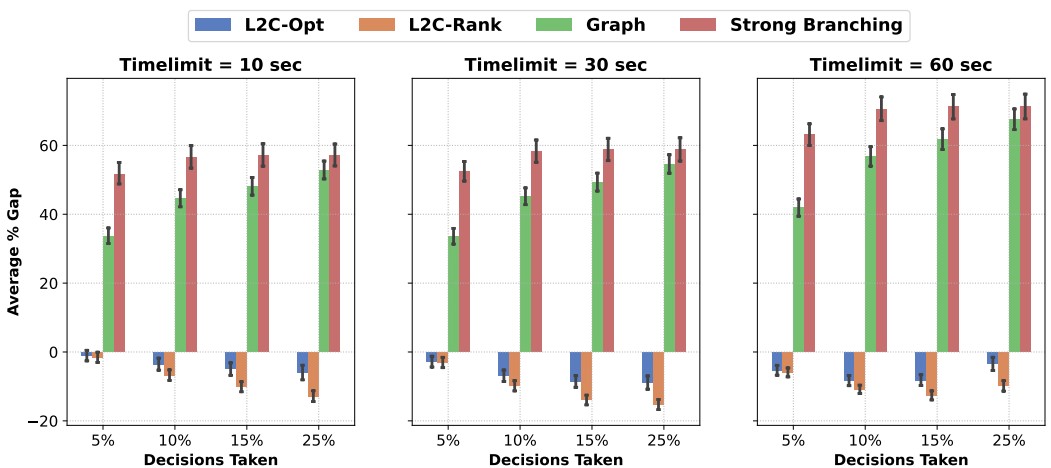

Figure 10: Average percentage gap on the BN 59 network for greedy conditioning using our methods, L2C-OPT and L2C-RANK, and baseline approaches across varying time budgets and numbers of decisions. More negative values indicate better performance.

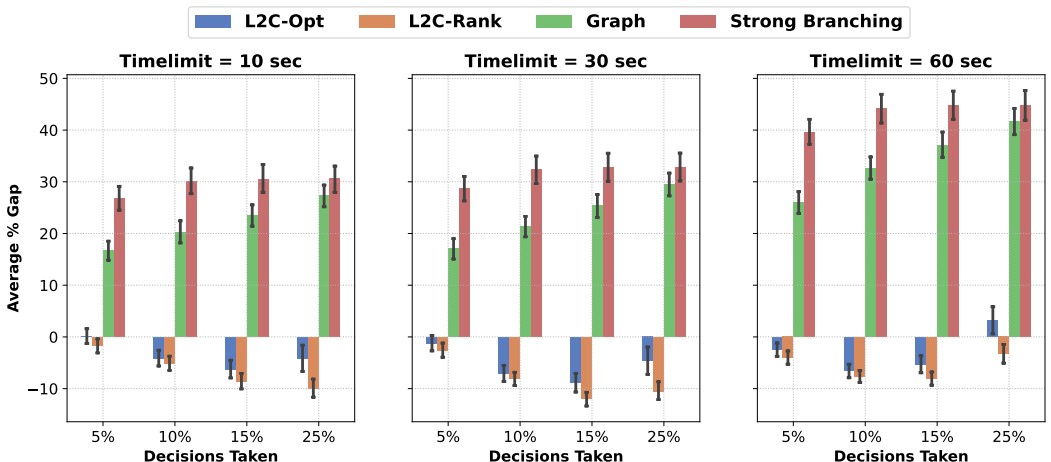

Figure 11: Average percentage gap on the BN 53 network for greedy conditioning using our methods, L2C-OPT and L2C-RANK, and baseline approaches across varying time budgets and numbers of decisions. More negative values indicate better performance.

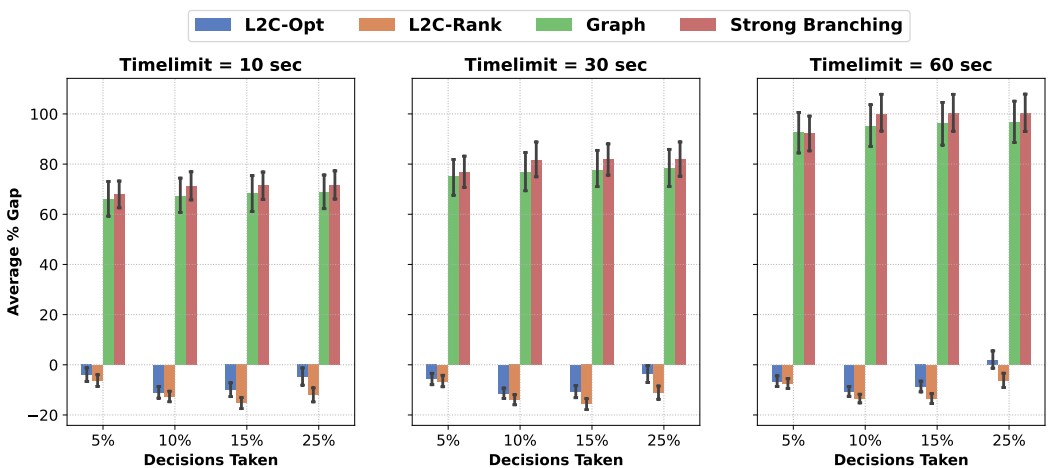

Figure 12: Average percentage gap on the BN 49 network for greedy conditioning using our methods, L2C-OPT and L2C-RANK, and baseline approaches across varying time budgets and numbers of decisions. More negative values indicate better performance.

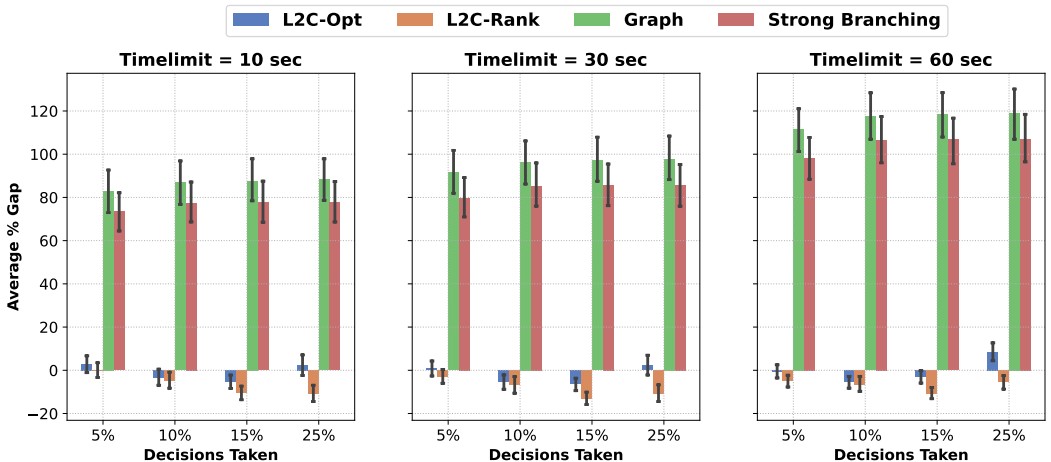

Figure 13: Average percentage gap on the BN 61 network for greedy conditioning using our methods, L2C-OPT and L2C-RANK, and baseline approaches across varying time budgets and numbers of decisions. More negative values indicate better performance.

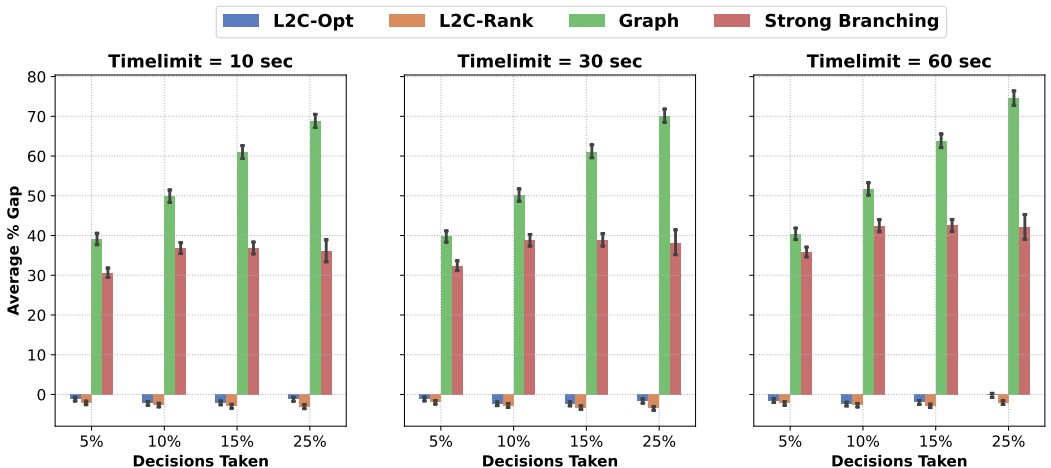

Figure 14: Average percentage gap on the BN 45 network for greedy conditioning using our methods, L2C-OPT and L2C-RANK, and baseline approaches across varying time budgets and numbers of decisions. More negative values indicate better performance.

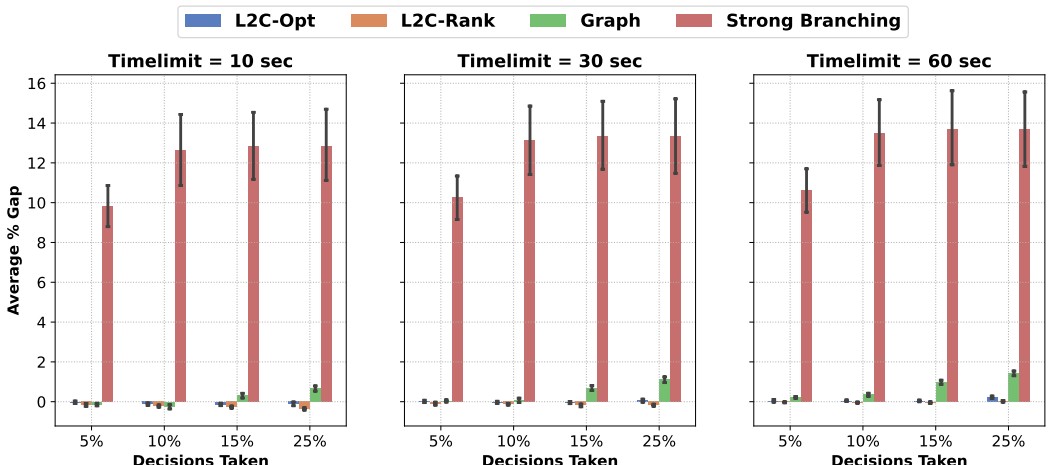

Figure 15: Average percentage gap on the Promedas 68 network for greedy conditioning using our methods, L2C-OPT and L2C-RANK, and baseline approaches across varying time budgets and numbers of decisions. More negative values indicate better performance.

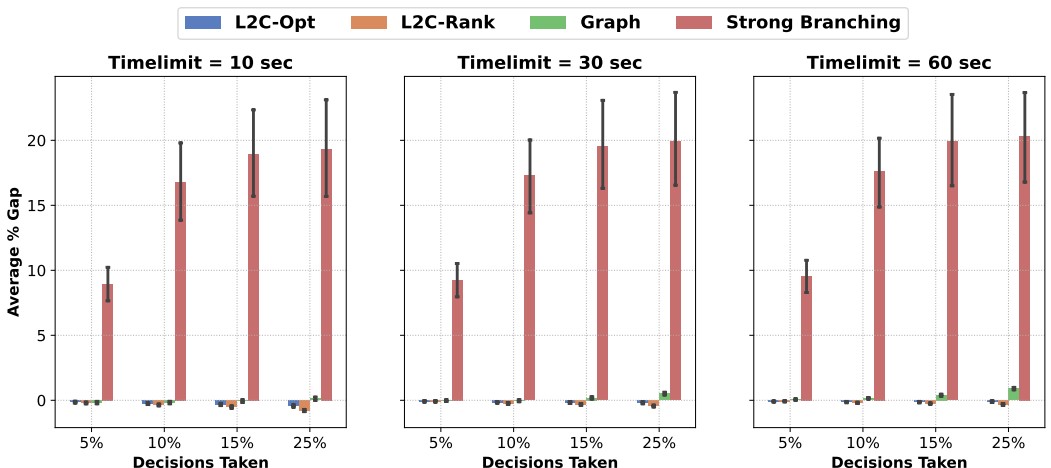

Figure 16: Average percentage gap on the Promedas 60 network for greedy conditioning using our methods, L2C-OPT and L2C-RANK, and baseline approaches across varying time budgets and numbers of decisions. More negative values indicate better performance.

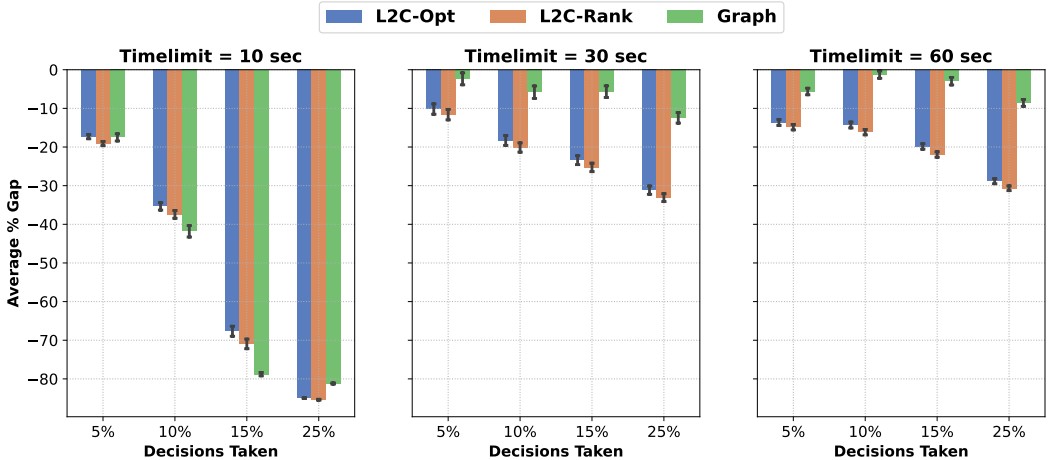

Figure 17: Average percentage gap on the BN 30 network for greedy conditioning using our methods, L2C-OPT and L2C-RANK, and baseline approaches across varying time budgets and numbers of decisions. More negative values indicate better performance.

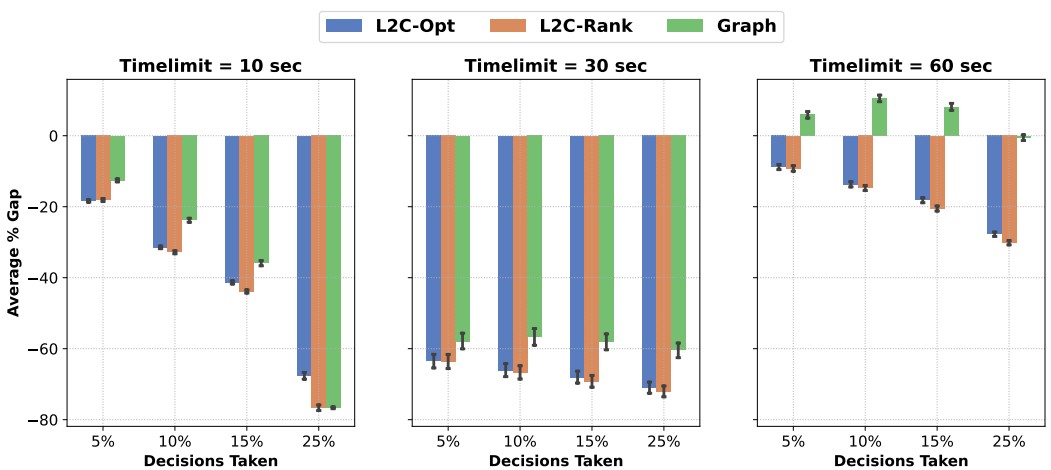

Figure 18: Average percentage gap on the BN 32 network for greedy conditioning using our methods, L2C-OPT and L2C-RANK, and baseline approaches across varying time budgets and numbers of decisions. More negative values indicate better performance.

# E    Conditioning performance for beam search-based conditioning

## E.1    Using SCIP as oracle

We now present detailed results comparing the performance of various conditioning strategies when used in beam search, using the SCIP solver as the oracle. Each bar represents the average log-likelihood gap (computed using equation 1) before and after conditioning, aggregated over conditioning depths of 5%, 10%, 15%, and 25% of the query variables, and averaged across all MPE queries solved using the oracle. Each subfigure corresponds to a specific oracle time budget.

Our neural strategies, L2C-OPT and L2C-RANK, consistently outperform the **full strong branching heuristic** in improving oracle performance, as evidenced by the negative average percentage gap. As the beam width increases, the performance of our methods either improves or remains consistent. In contrast, the strong branching heuristic often fails to return a decision within the 30-second time limit on larger networks and wider beams. As a result, the corresponding bars are omitted in the plots.

We omit results for the **graph-based heuristic**, as it only supports a beam width of 1 and is therefore not applicable in this setting.

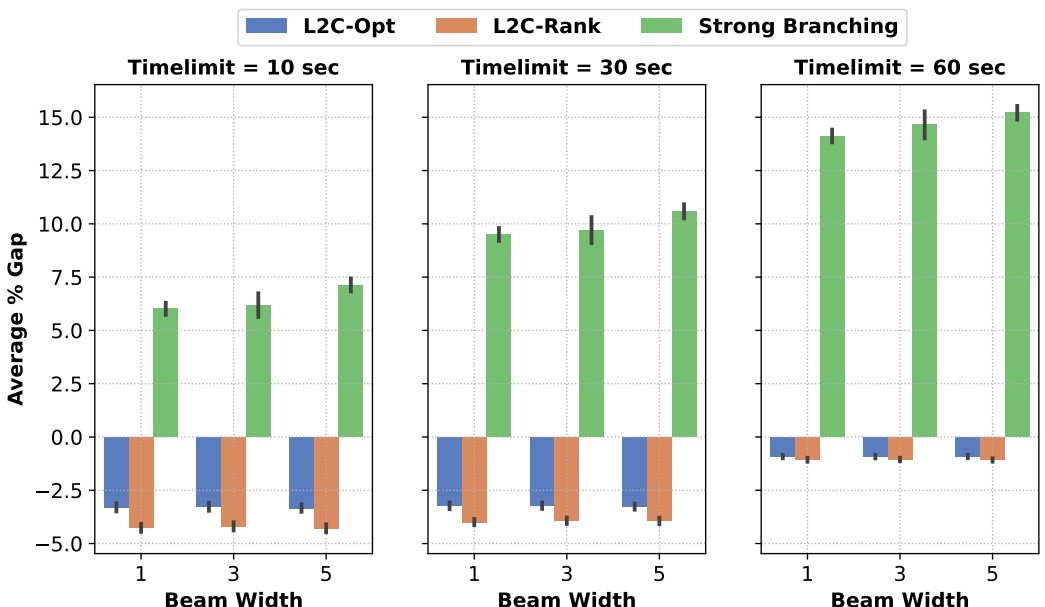

Figure 19: Average percentage gap on the BN 12 network comparing our methods to baseline approaches across varying time budgets and beam widths using beam-search based conditioning. More negative values indicate better performance.

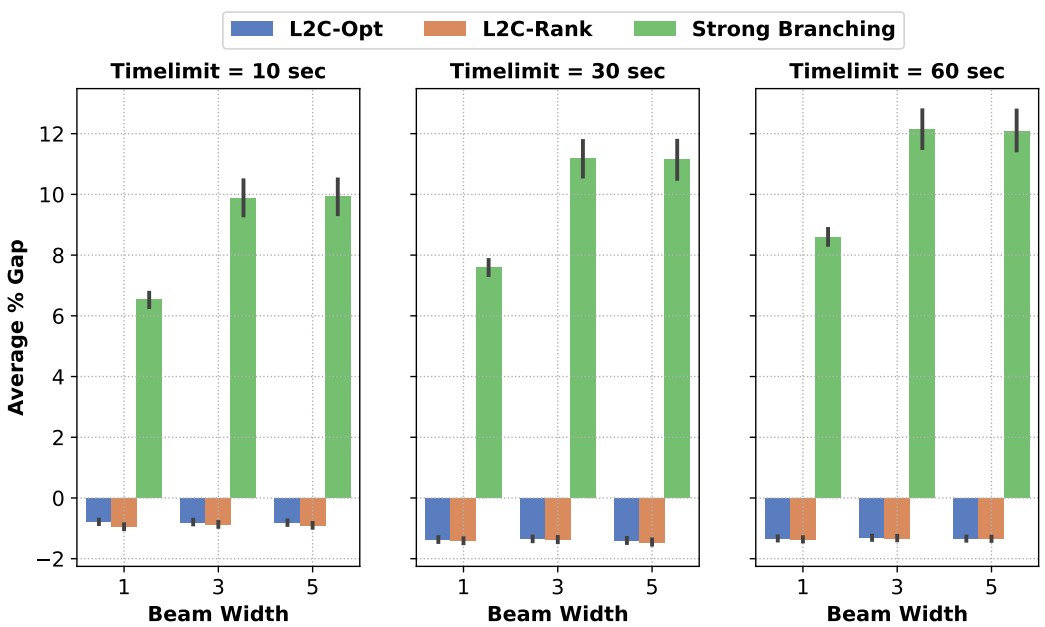

Figure 20: Average percentage gap on the BN 9 network comparing our methods to baseline approaches across varying time budgets and beam widths using beam-search based conditioning. More negative values indicate better performance.

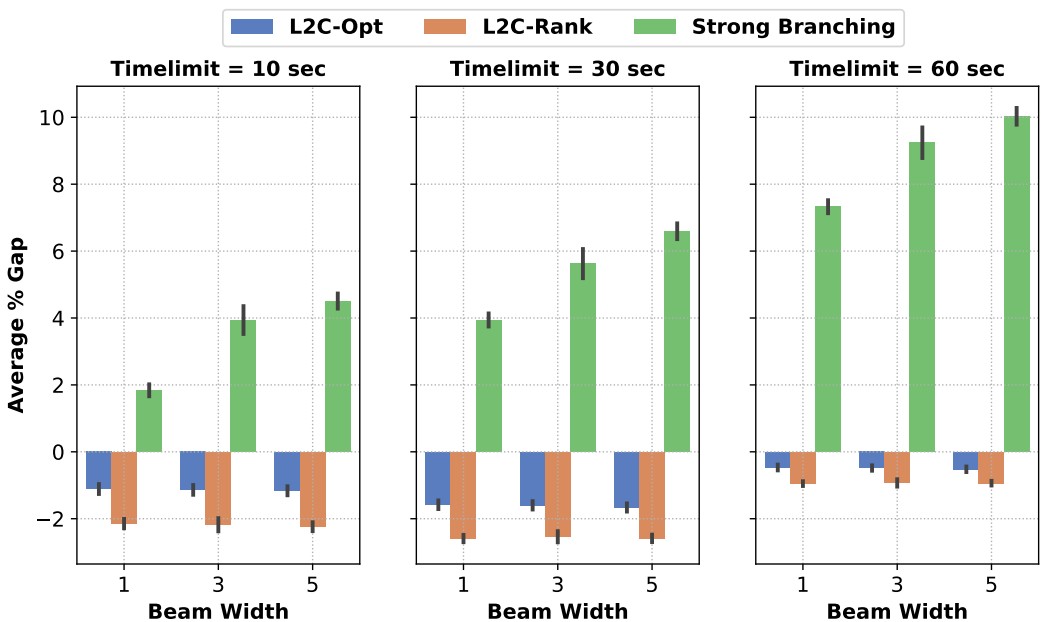

Figure 21: Average percentage gap on the BN 13 network comparing our methods to baseline approaches across varying time budgets and beam widths using beam-search based conditioning. More negative values indicate better performance.

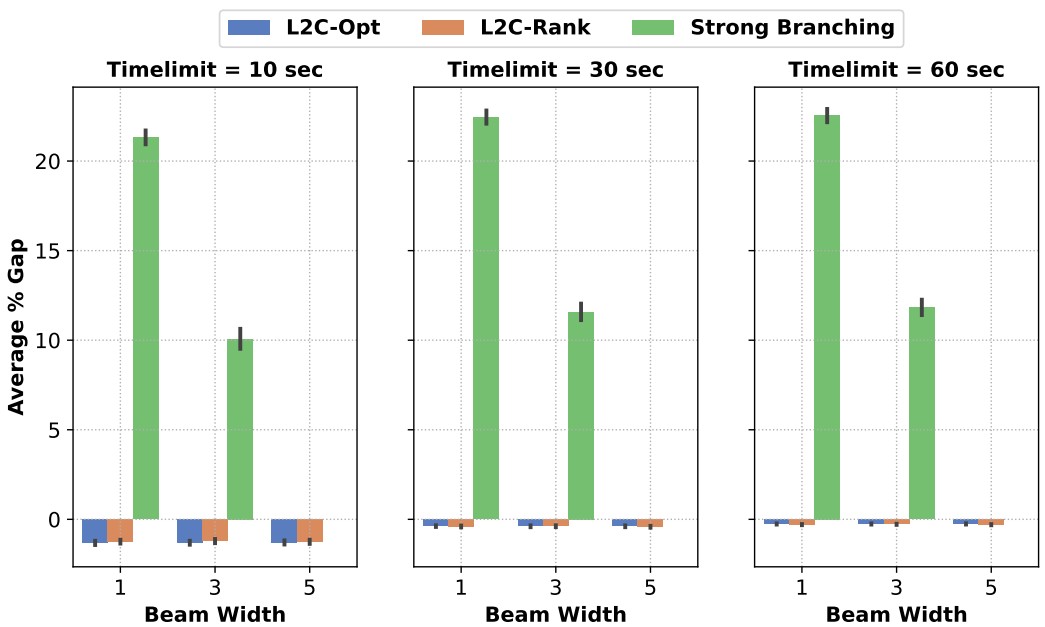

Figure 22: Average percentage gap on the Grid 20 network comparing our methods to baseline approaches across varying time budgets and beam widths using beam-search based conditioning. More negative values indicate better performance. Missing bars for strong branching indicate timeouts.

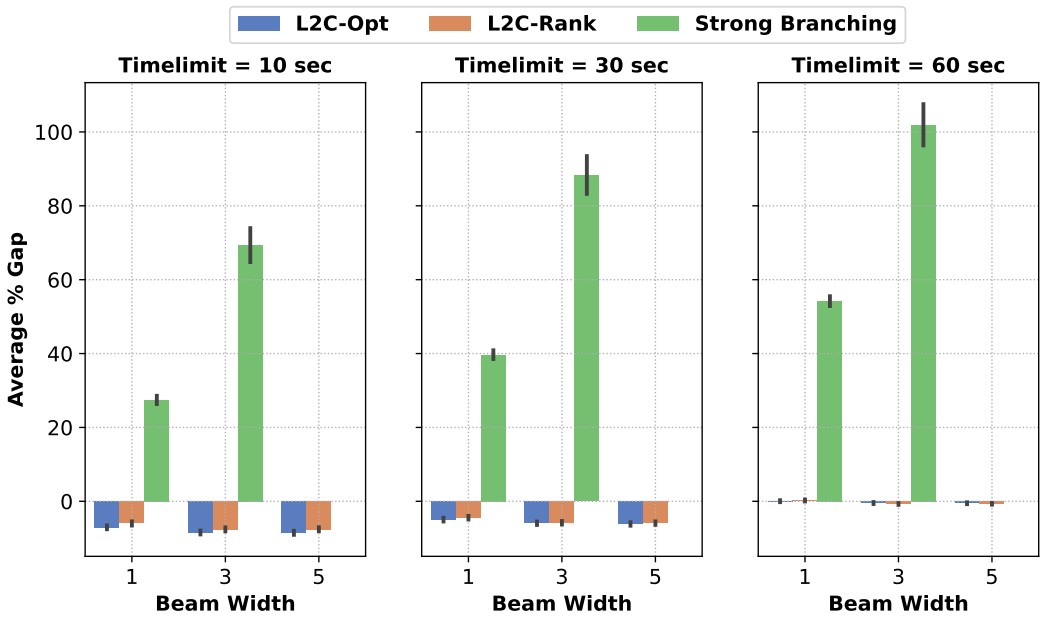

Figure 23: Average percentage gap on the BN 65 network comparing our methods to baseline approaches across varying time budgets and beam widths using beam-search based conditioning. More negative values indicate better performance. Missing bars for strong branching indicate timeouts.

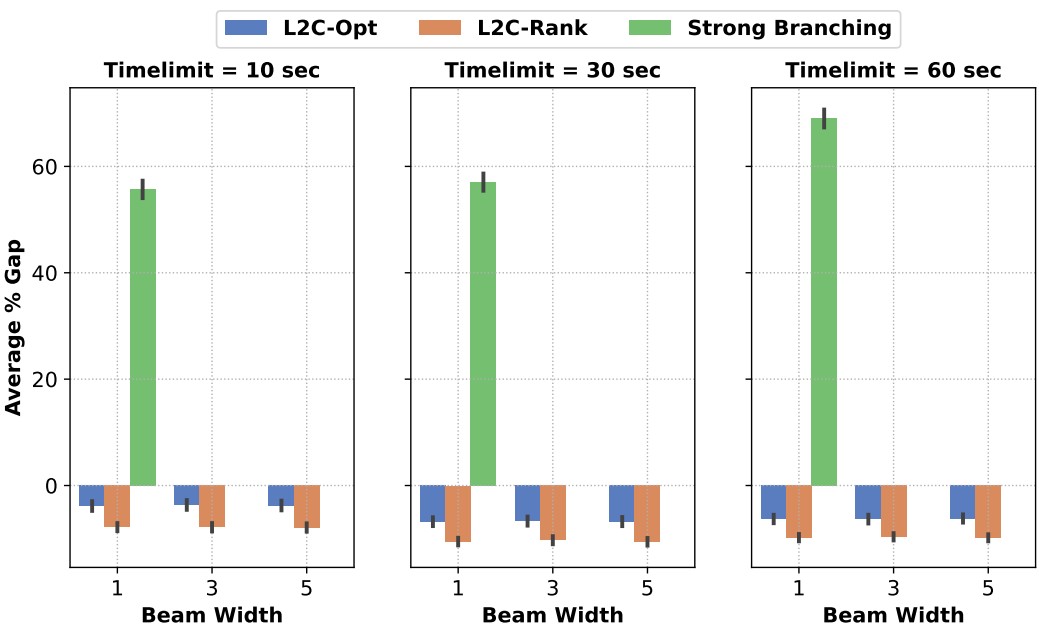

Figure 24: Average percentage gap on the BN 59 network comparing our methods to baseline approaches across varying time budgets and beam widths using beam-search based conditioning. More negative values indicate better performance. Missing bars for strong branching indicate timeouts.

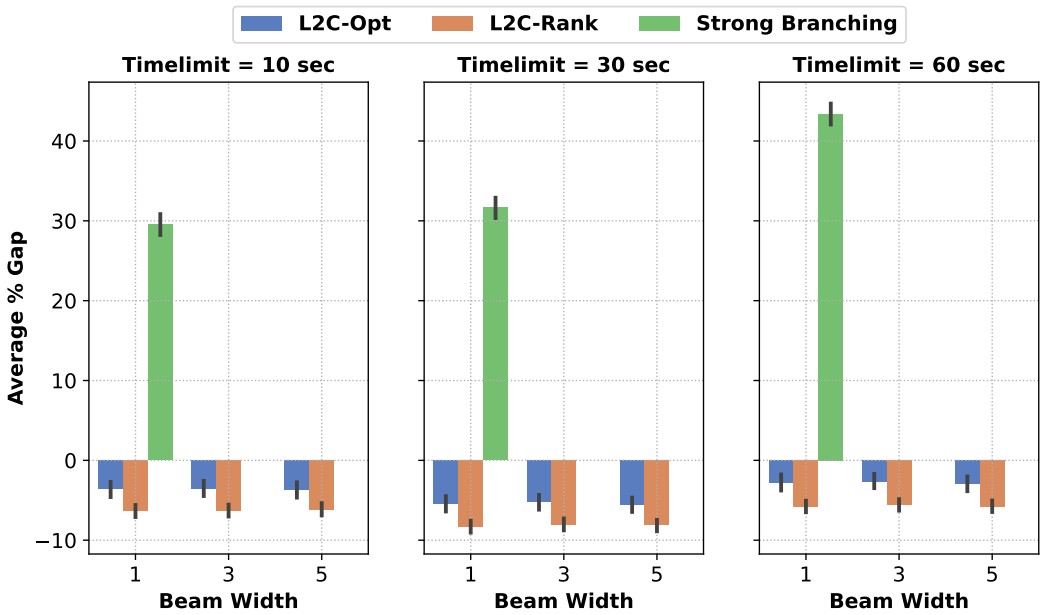

Figure 25: Average percentage gap on the BN 53 network comparing our methods to baseline approaches across varying time budgets and beam widths using beam-search based conditioning. More negative values indicate better performance. Missing bars for strong branching indicate timeouts.

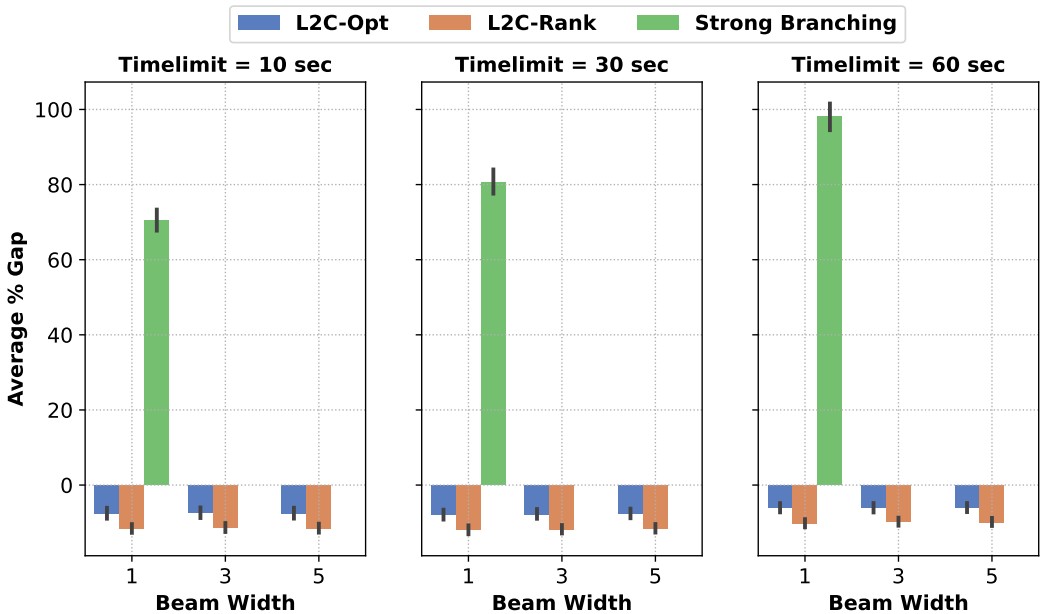

Figure 26: Average percentage gap on the BN 49 network comparing our methods to baseline approaches across varying time budgets and beam widths using beam-search based conditioning. More negative values indicate better performance. Missing bars for strong branching indicate timeouts.

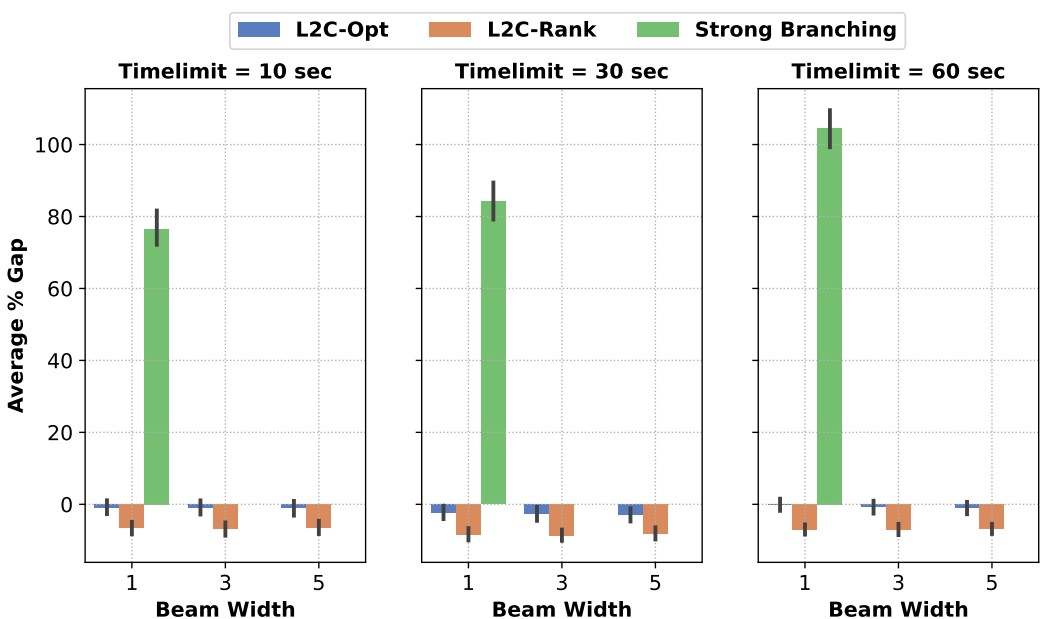

Figure 27: Average percentage gap on the BN 61 network comparing our methods to baseline approaches across varying time budgets and beam widths using beam-search based conditioning. More negative values indicate better performance. Missing bars for strong branching indicate timeouts.

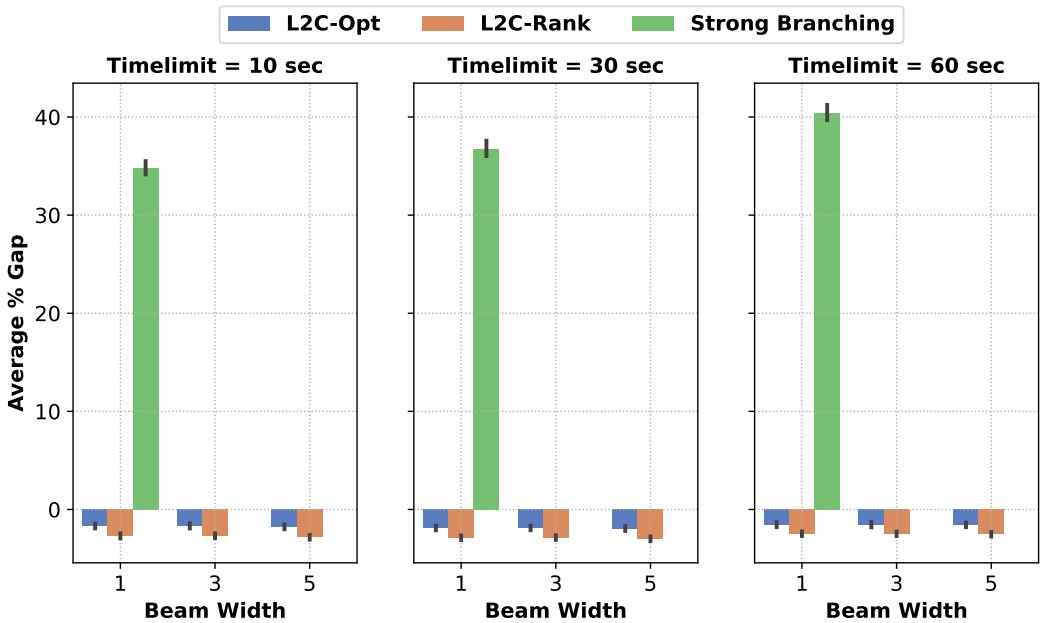

Figure 28: Average percentage gap on the BN 45 network comparing our methods to baseline approaches across varying time budgets and beam widths using beam-search based conditioning. More negative values indicate better performance. Missing bars for strong branching indicate timeouts.

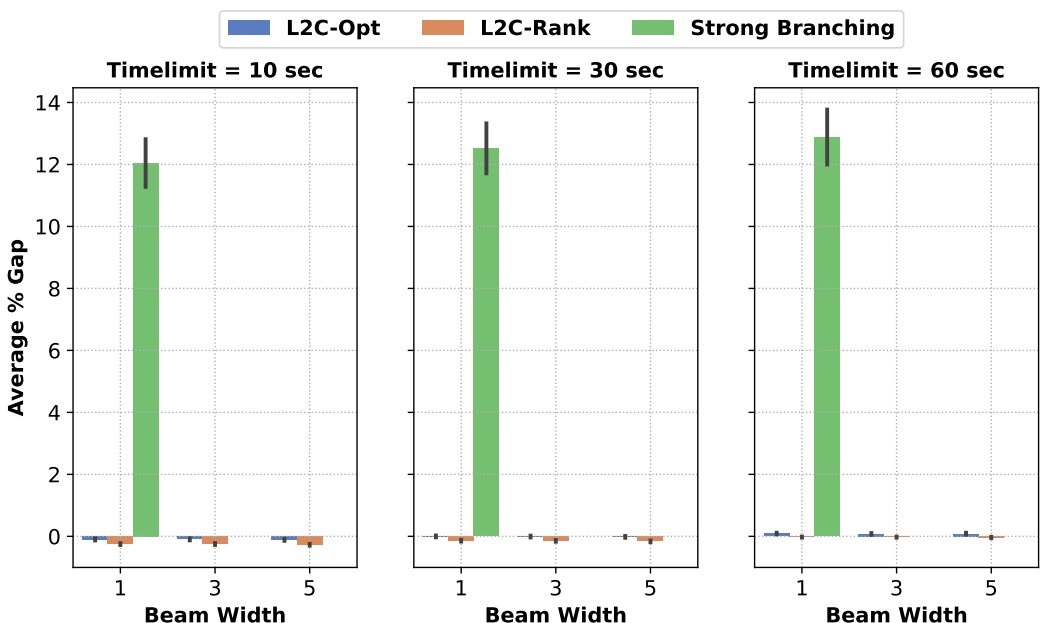

Figure 29: Average percentage gap on the Promedas 68 network comparing our methods to baseline approaches across varying time budgets and beam widths using beam-search based conditioning. More negative values indicate better performance. Missing bars for strong branching indicate timeouts.

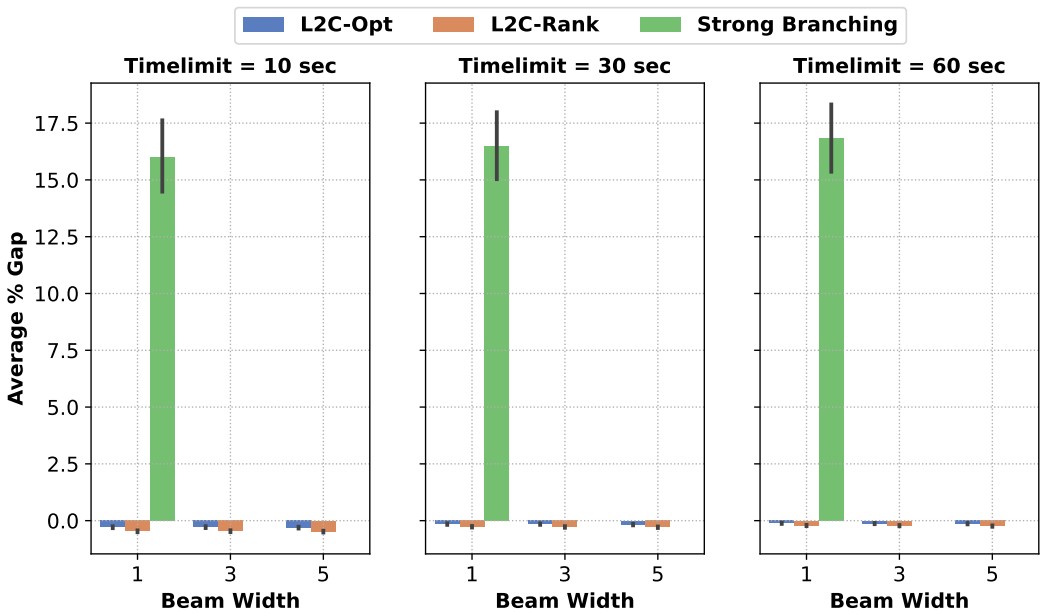

Figure 30: Average percentage gap on the Promedas 60 network comparing our methods to baseline approaches across varying time budgets and beam widths using beam-search based conditioning. More negative values indicate better performance. Missing bars for strong branching indicate timeouts.

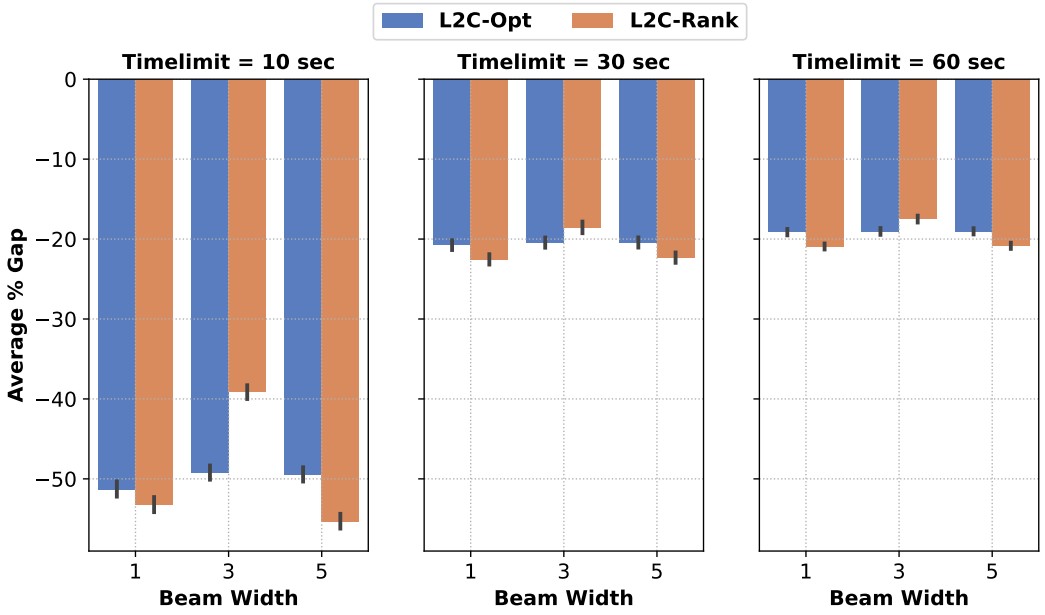

Figure 31: Average percentage gap on the BN 30 network comparing our methods to baseline approaches across varying time budgets and beam widths using beam-search based conditioning. More negative values indicate better performance. Missing bars for strong branching indicate timeouts.

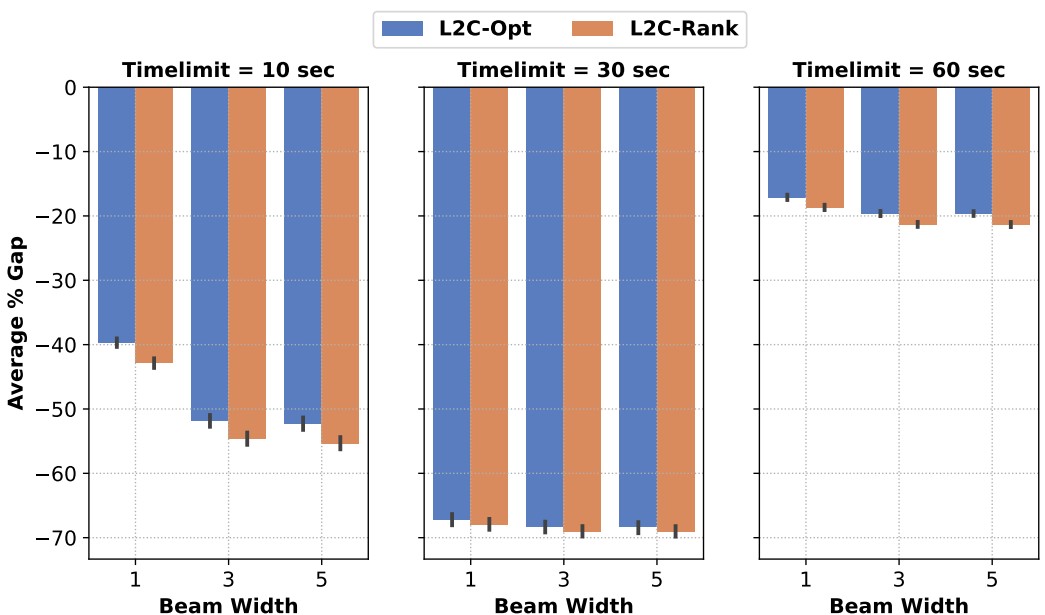

Figure 32: Average percentage gap on the BN 32 network comparing our methods to baseline approaches across varying time budgets and beam widths using beam-search based conditioning. More negative values indicate better performance. Missing bars for strong branching indicate timeouts.

## E.2 Using AOBB as oracle

In this section, we evaluate the performance of beam search-based conditioning with AOBB (per Otten and Dechter [24] and implemented by Otten [3]) as the oracle, using beam widths of 3 and 5, as shown in Figures 33 and 34, respectively. Each figure plots the average solution gap against the percentage reduction in the number of search nodes across 1,000 MPE queries over all networks, with each point representing one network. The solution gap is computed using Equation 1, while node reduction is calculated using the same formula with log-likelihood scores replaced by node counts before and after conditioning.

Our methods, L2C-OPT and L2C-RANK, consistently yield lower solution gapsindicating better preservation of optimal solutionsand higher node reductions, reflecting improved search efficiency. In contrast, the full strong branching heuristic frequently fails to preserve solution quality, as indicated by its larger gaps, and produces fewer data points due to timeouts exceeding the 30-second limit per decision.

As before, we omit results for the **graph-based heuristic**, as it supports only a beam width of 1 and is therefore not applicable in this setting.

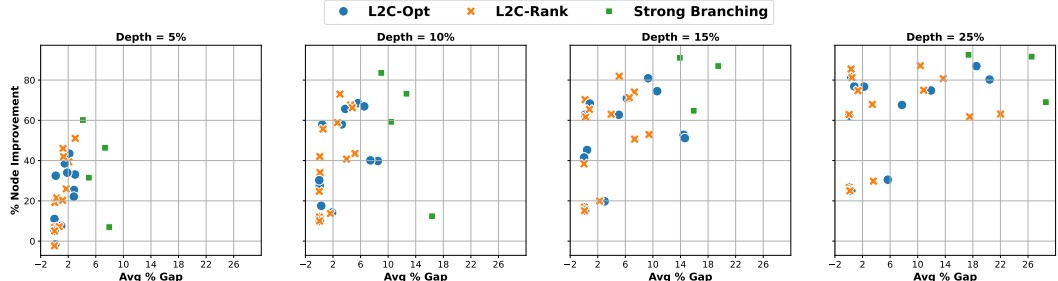

Figure 33: Beam search-based conditioning with AOBB as the oracle using a beam width of 3. Each subfigure corresponds to a fixed number of conditioning decisions. The x-axis indicates the average solution gap (lower is better), and the y-axis indicates the percentage reduction in node count (higher is better). Each point represents a single network.

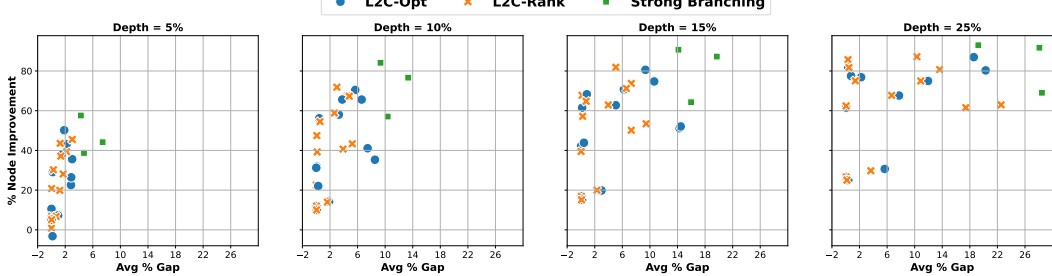

Figure 34: Beam search-based conditioning with AOBB as the oracle using a beam width of 5. Each subfigure corresponds to a fixed number of conditioning decisions. The x-axis indicates the average solution gap (lower is better), and the y-axis indicates the percentage reduction in node count (higher is better). Each point represents a single network.

## F  Incorporating L2C scores as branching and node selection heuristics in branch-and-bound

In this section, we compare our trained neural networksused as branching rules [32] and node selection heuristics [34] within the SCIP frameworkwith SCIPs default heuristics [23, 49, 50]. We evaluate performance based on the average percentage gap in log-likelihood (LL) scores between our methods, L2C-OPT and L2C-RANK, and SCIPs default strategies on the same set of MPE queries. The gap is computed as:

$$\frac{1}{N} \sum_{i=1}^{N} \frac{\mathcal{LL}_S^{(i)} - \mathcal{LL}_N^{(i)}}{|\mathcal{LL}_S^{(i)}|} \times 100 \qquad (2)$$

where $\mathcal{LL}_S^{(i)}$ denotes the log-likelihood score achieved by SCIP's default heuristics, and $\mathcal{LL}_N^{(i)}$ denotes the score obtained using our L2C methods on the $i$-th instance. Negative values indicate that our methods perform better; positive values indicate superior performance by SCIPs default heuristics.

As shown in Figures 35 to 48, the percentage gap is typically negative, demonstrating that our methods consistently yield higher log-likelihood scores than SCIP within the same time budget. This indicates that for time-constrained settings, L2C-OPT and L2C-RANK can find higher-quality solutions more efficiently than SCIPs default branching and node selection strategies. Overall, our learned heuristics not only produce better decisions but also execute faster than the state-of-the-art methods implemented in SCIP.

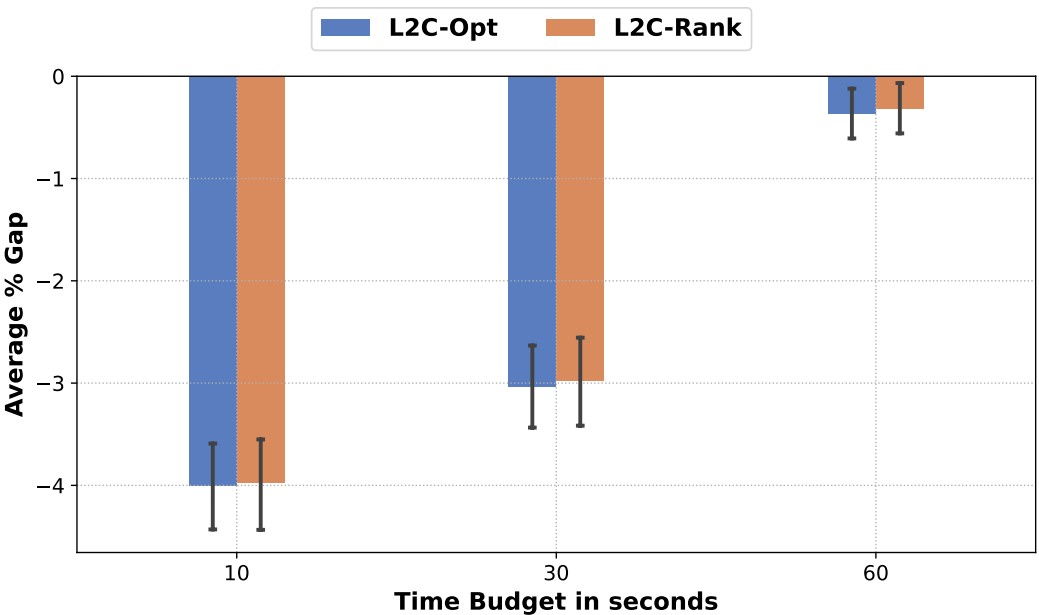

Figure 35: Comparison of SCIPs default heuristics with our neural strategies, L2C-OPT and L2C-RANK, for branching and node selection within the SCIP framework on the BN 12 network in terms of average % gap in log-likelihood. More negative values indicate better performance.

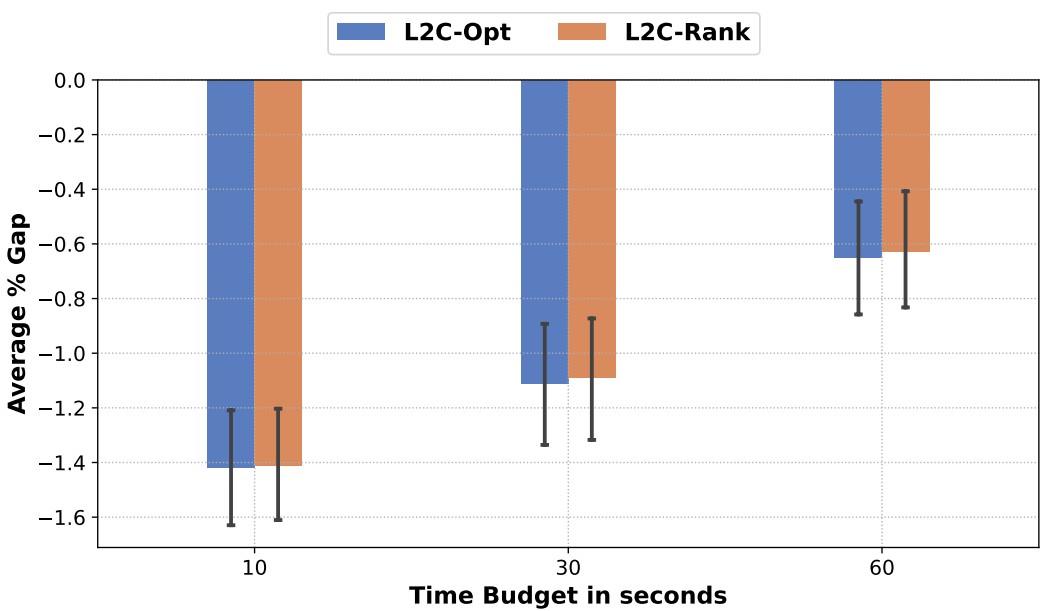

Figure 36: Comparison of SCIPs default heuristics with our neural strategies, L2C-OPT and L2C-RANK, for branching and node selection within the SCIP framework on the BN 9 network in terms of average % gap in log-likelihood. More negative values indicate better performance.

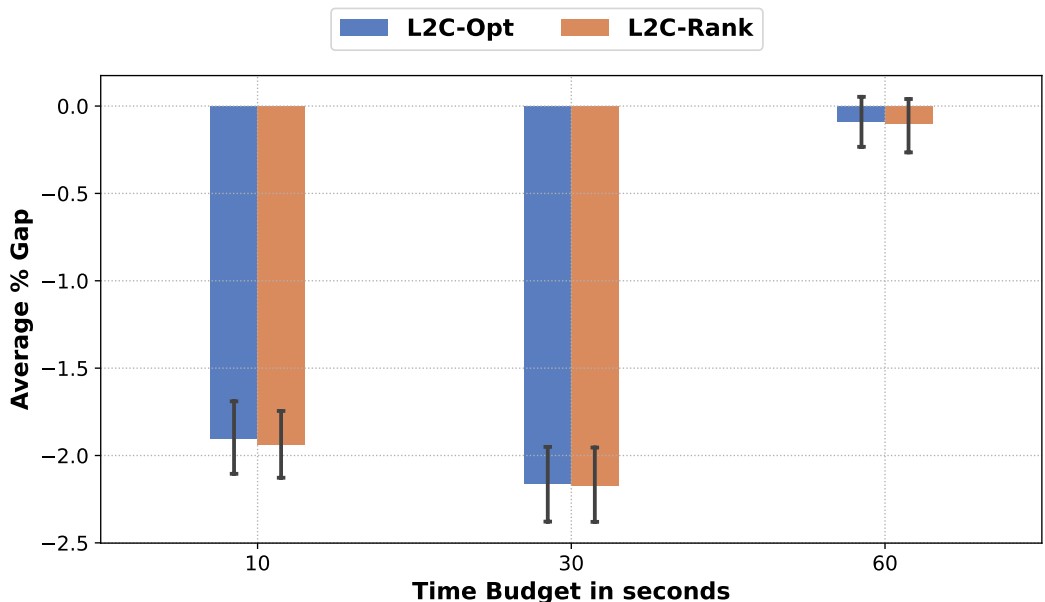

Figure 37: Comparison of SCIPs default heuristics with our neural strategies, L2C-OPT and L2C-RANK, for branching and node selection within the SCIP framework on the BN 13 network in terms of average % gap in log-likelihood. More negative values indicate better performance.

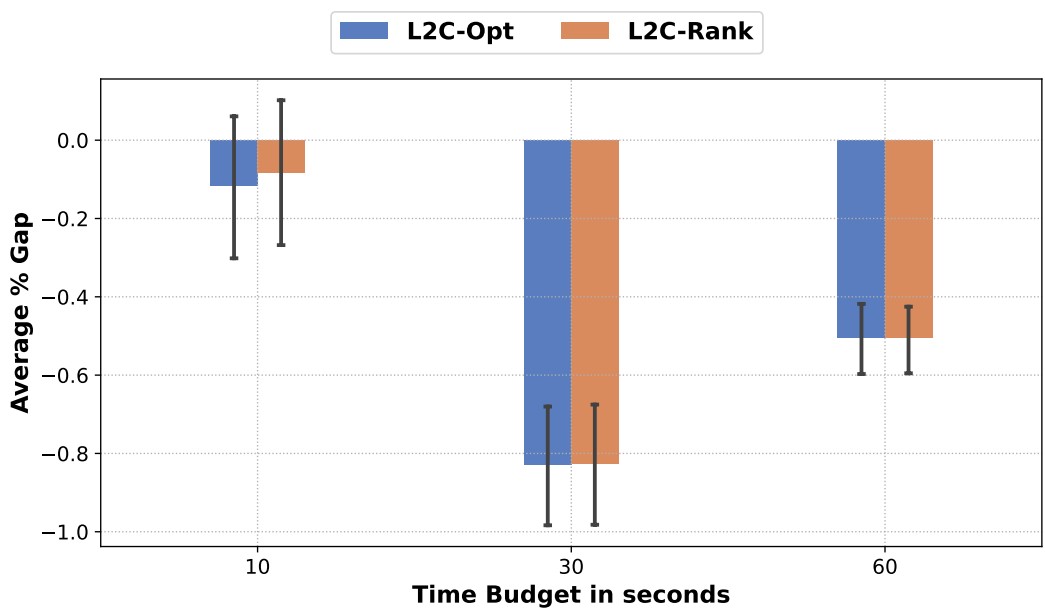

Figure 38: Comparison of SCIPs default heuristics with our neural strategies, L2C-OPT and L2C-RANK, for branching and node selection within the SCIP framework on the Grid 20 network in terms of average % gap in log-likelihood. More negative values indicate better performance.

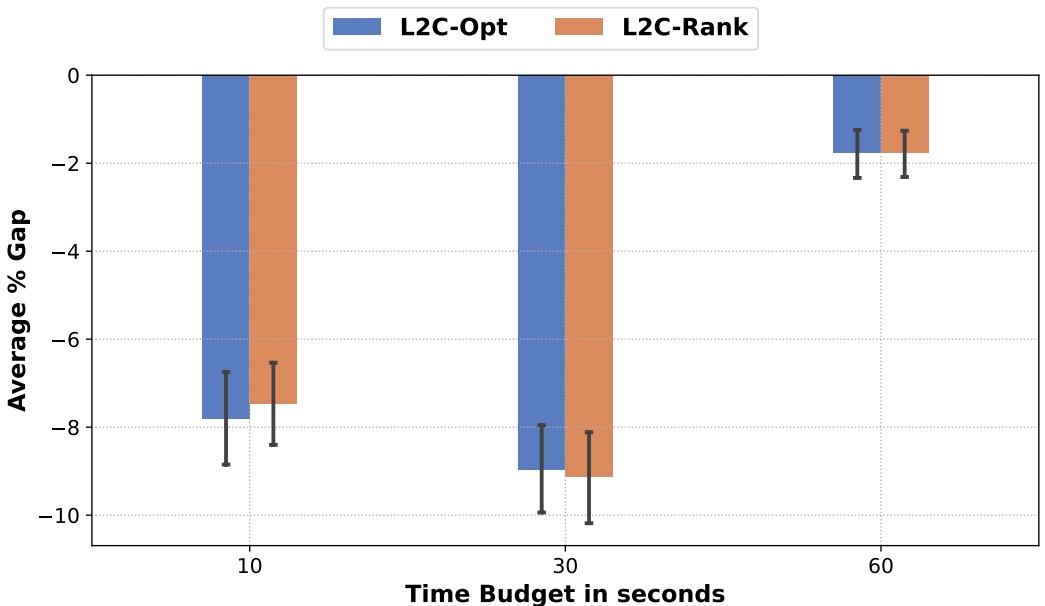

Figure 39: Comparison of SCIPs default heuristics with our neural strategies, L2C-OPT and L2C-RANK, for branching and node selection within the SCIP framework on the BN 65 network in terms of average % gap in log-likelihood. More negative values indicate better performance.

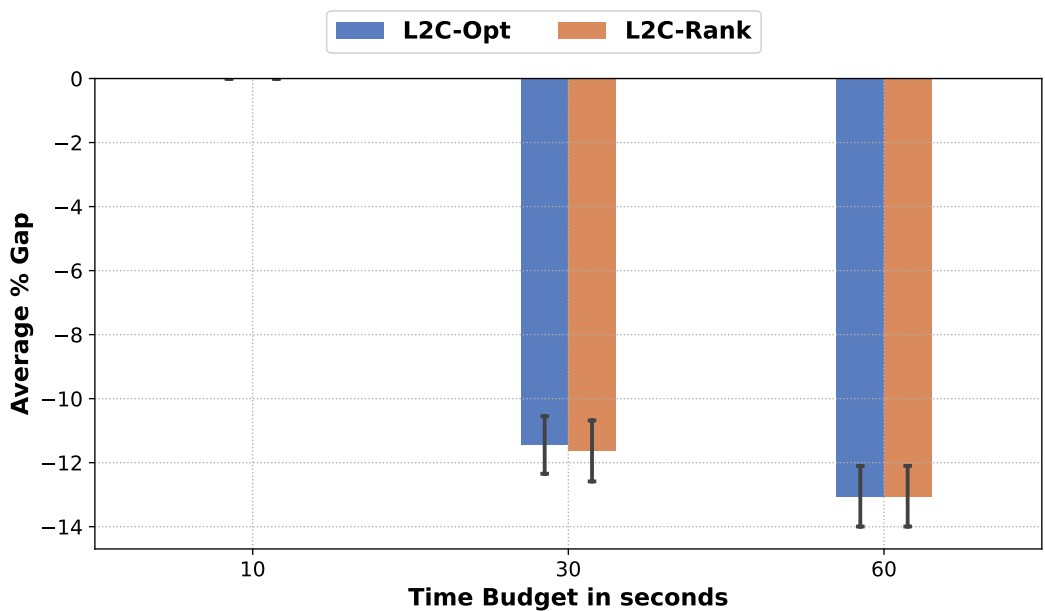

Figure 40: Comparison of SCIPs default heuristics with our neural strategies, L2C-OPT and L2C-RANK, for branching and node selection within the SCIP framework on the BN 59 network in terms of average % gap in log-likelihood. More negative values indicate better performance.

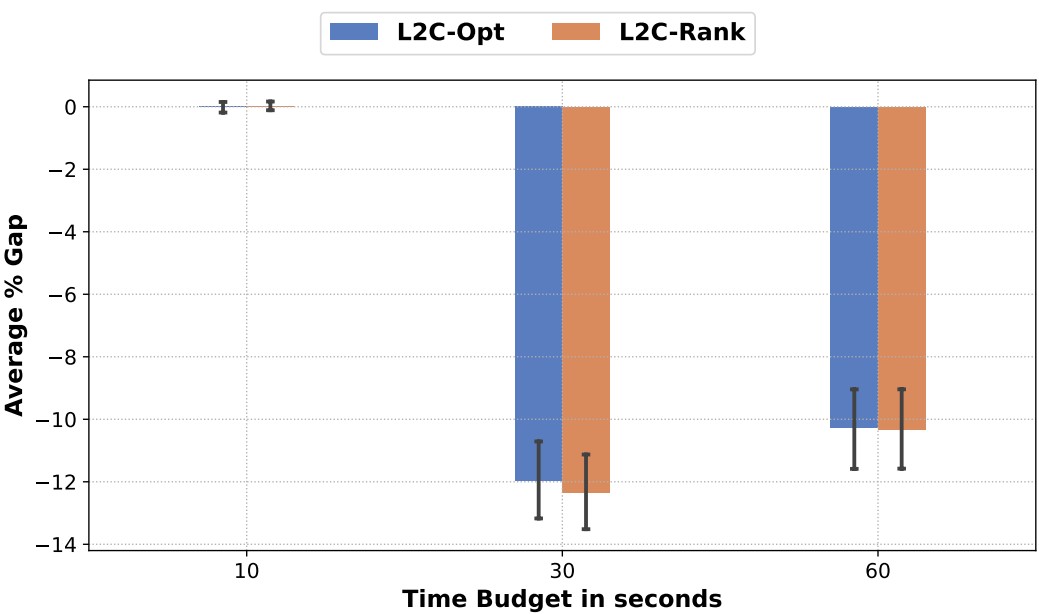

Figure 41: Comparison of SCIPs default heuristics with our neural strategies, L2C-OPT and L2C-RANK, for branching and node selection within the SCIP framework on the BN 53 network in terms of average % gap in log-likelihood. More negative values indicate better performance.

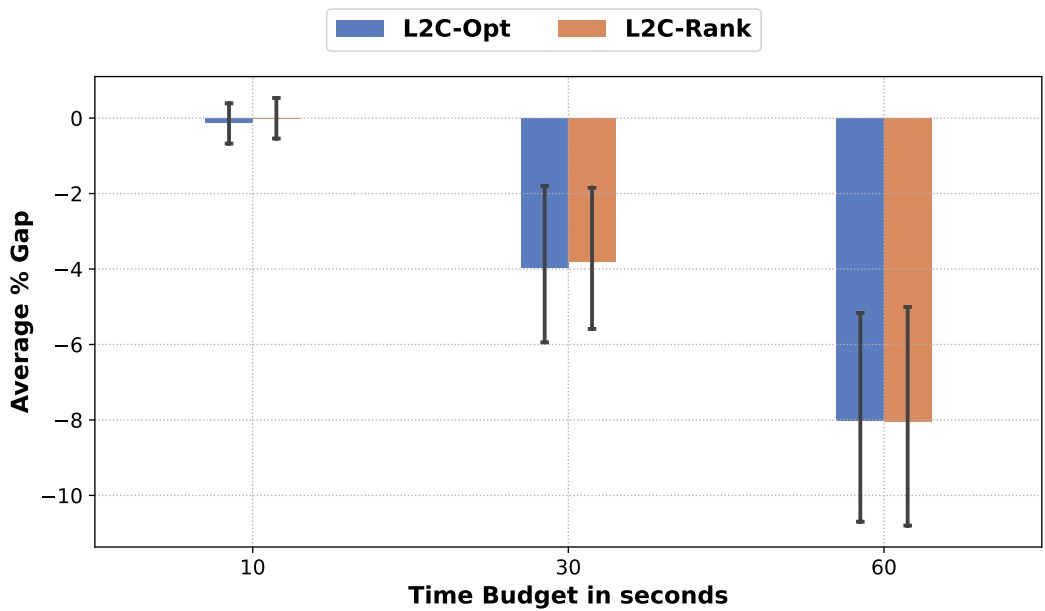

Figure 42: Comparison of SCIPs default heuristics with our neural strategies, L2C-OPT and L2C-RANK, for branching and node selection within the SCIP framework on the BN 49 network in terms of average % gap in log-likelihood. More negative values indicate better performance.

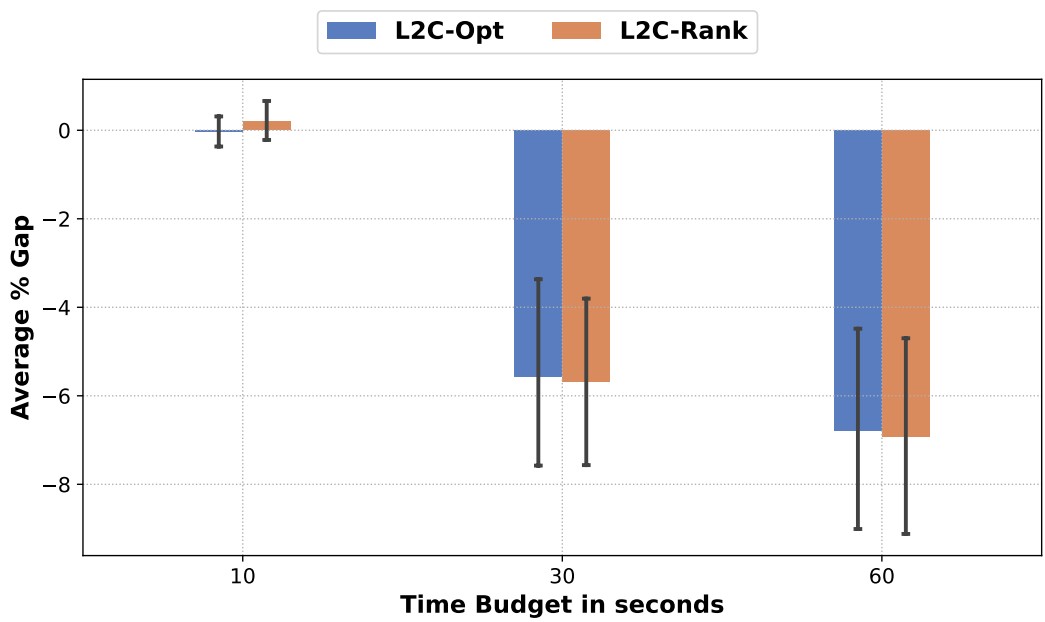

Figure 43: Comparison of SCIPs default heuristics with our neural strategies, L2C-OPT and L2C-RANK, for branching and node selection within the SCIP framework on the BN 61 network in terms of average % gap in log-likelihood. More negative values indicate better performance.

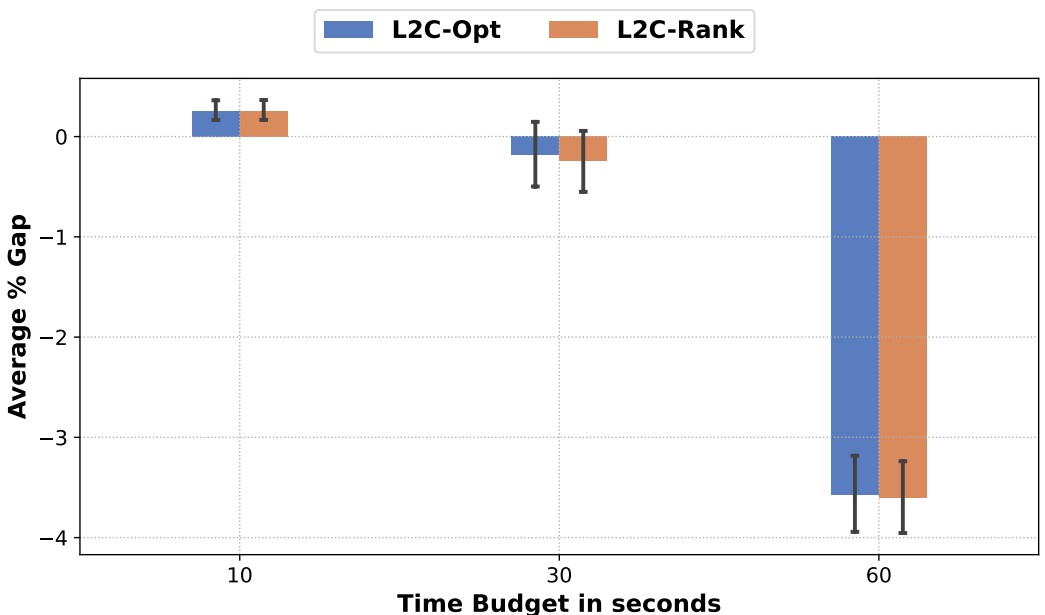

Figure 44: Comparison of SCIPs default heuristics with our neural strategies, L2C-OPT and L2C-RANK, for branching and node selection within the SCIP framework on the BN 45 network in terms of average % gap in log-likelihood. More negative values indicate better performance.

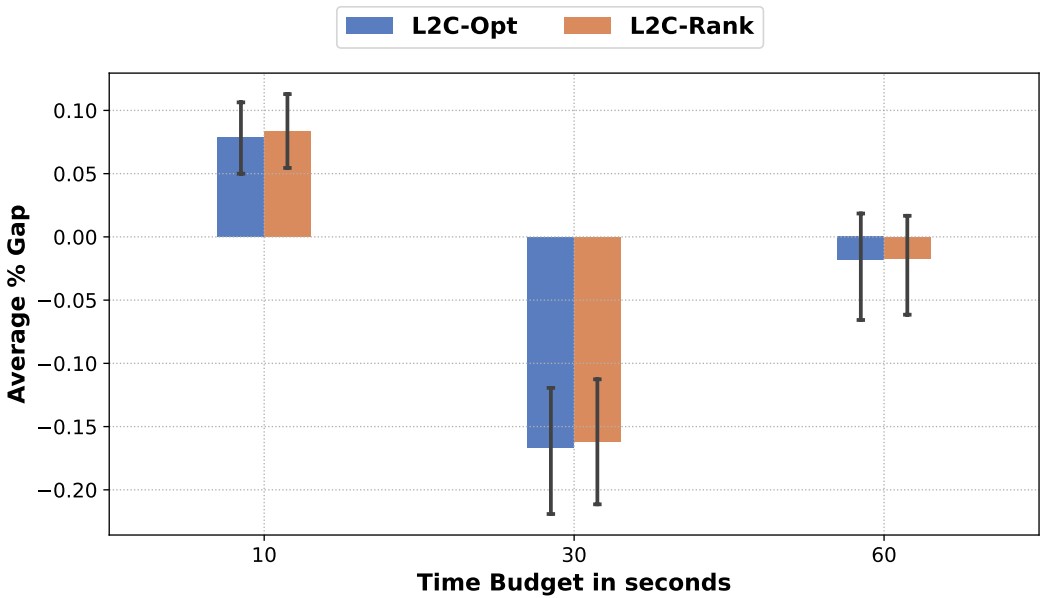

Figure 45: Comparison of SCIPs default heuristics with our neural strategies, L2C-OPT and L2C-RANK, for branching and node selection within the SCIP framework on the Promedas 68 network in terms of average % gap in log-likelihood. More negative values indicate better performance.

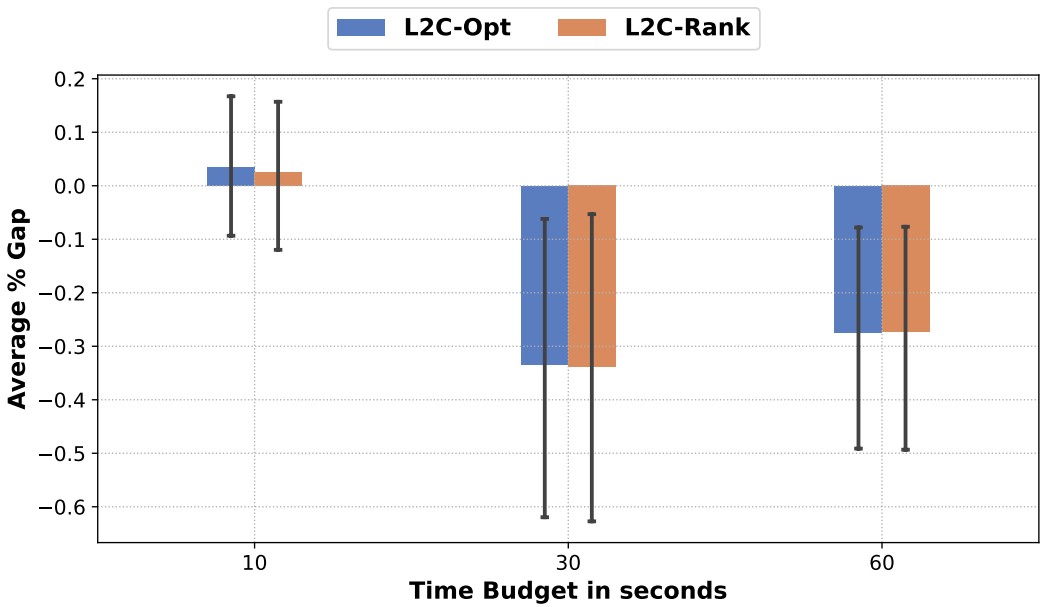

Figure 46: Comparison of SCIPs default heuristics with our neural strategies, L2C-OPT and L2C-RANK, for branching and node selection within the SCIP framework on the Promedas 60 network in terms of average % gap in log-likelihood. More negative values indicate better performance.

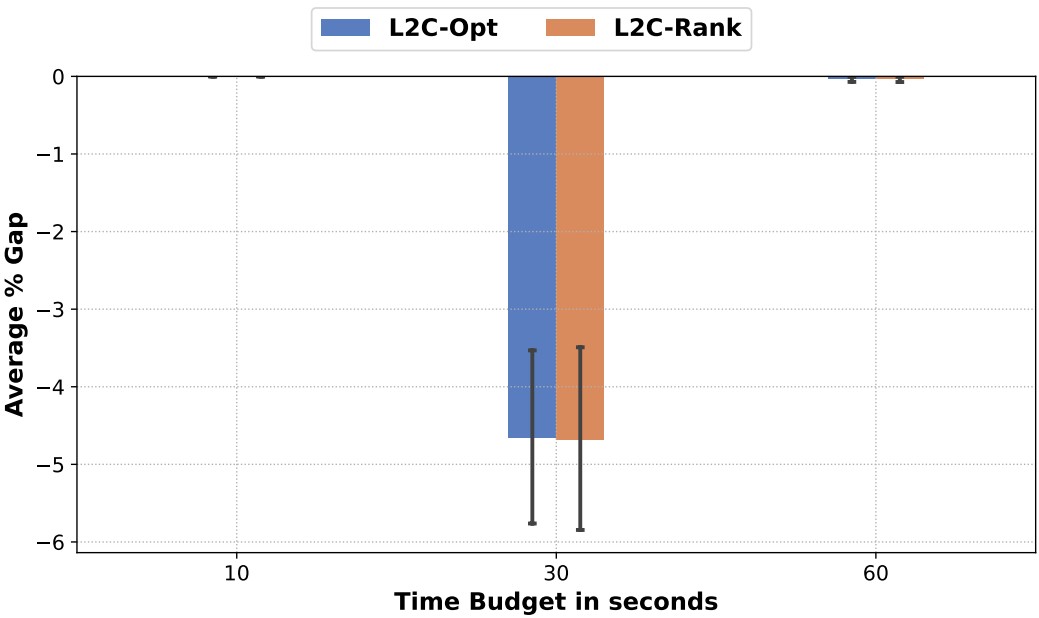

Figure 47: Comparison of SCIPs default heuristics with our neural strategies, L2C-OPT and L2C-RANK, for branching and node selection within the SCIP framework on the BN 30 network in terms of average % gap in log-likelihood. More negative values indicate better performance.

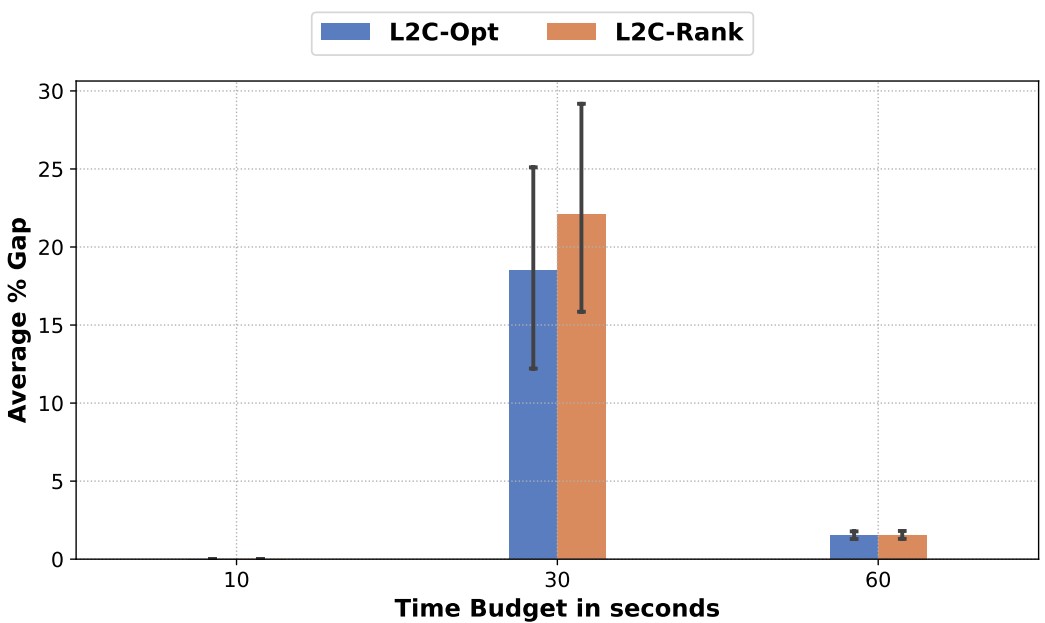

Figure 48: Comparison of SCIPs default heuristics with our neural strategies, L2C-Opt and L2C-Rank, for branching and node selection within the SCIP framework on the BN 32 network in terms of average % gap in log-likelihood. More negative values indicate better performance.

