# OpenReview forum: "Learning to Condition: A Neural Heuristic for Scalable MPE Inference"
_NeurIPS.cc/2025/Conference — NeurIPS 2025 poster_

### Official Review · Reviewer_GK5K · 2025-06-29

**Clarity:** 3
**Significance:** 2
**Originality:** 3
**Rating:** 4
**Confidence:** 3

**Summary:**

This paper introduces Learning to Condition (L2C), a neural heuristic framework for scalable Most Probable Explanation (MPE) inference in Probabilistic Graphical Models (PGMs). The core idea is to train an attention-based neural network to score variable-value assignments based on their utility for conditioning, balancing the preservation of optimal solutions with the simplification of inference. The model is supervised using a scalable data pipeline that extracts training signals from the search traces of existing MPE solvers. L2C is integrated with search algorithms, serving as a conditioning strategy before exact inference or as a branching and node selection policy within branch-and-bound solvers. Experiments on high-treewidth PGMs demonstrate that L2C significantly reduces the search space while maintaining or improving solution quality compared to state-of-the-art methods.

**Questions:**

- The conditioning problem resembles MAXSAT (weighted constraint optimization). How does L2C compare to state-of-the-art PB solvers (e.g., using cutting planes or conflict analysis) in efficiency?

**Ethical Concerns:**

["NO or VERY MINOR ethics concerns only"]

**Final Justification:**

Thank you for your response, which clarified my questions. My concerns are resolved, so I maintain my positive score of 4.

**Limitations:**

Yes

**Quality:**

3

**Strengths And Weaknesses:**

Strengths:
- Innovative Integration: Combines neural networks with symbolic reasoning, using solver-guided feedback to ensure semantic correctness of generated constraints.
- Data Efficiency: The supervision strategy leverages solver statistics without requiring exhaustive MPE enumeration.
- Flexible Deployment: Functions as both a pre-processing conditioning strategy and a branching heuristic within branch-and-bound solvers.
Weaknesses:
- Binary Limitation: Assumes all variables are binary ("w.l.o.g.", Sec. 2). This restricts applicability to real-valued variables, limiting generalization beyond discrete domains.
- MAXSAT/PB Solver Comparison: Lacks direct comparison with pseudo-Boolean (PB) solvers, missing an opportunity to contextualize novelty against established methods.

---

> ### Author Rebuttal · Authors · 2025-07-30
>
> We sincerely appreciate Reviewer GK5K's thoughtful comments and constructive feedback. Below, we provide detailed responses to each of their points.
>
> ## Comment 1
>
> > Binary Limitation: Assumes all variables are binary ("w.l.o.g.", Sec. 2). This restricts applicability to real-valued variables, limiting generalization beyond discrete domains.
>
> ### Response
> Our proposed L2C method can be easily extended to multi-valued discrete variables and, with additional modifications, to continuous variables. We provide more details below.
>
> 1. **Extension to multi-valued discrete variables:**
>
> * **Data Collection:** The procedure remains unchanged.
>
> * **Training:** As in the binary case, we initialize embeddings for all variable–value pairs. Our attention-based architecture naturally supports multi-valued variables, as it outputs scores for each pair independently. The optimality and simplification heads produce per-pair scores, and the existing loss functions remain applicable without modification.
>
> * **Inference:** The same sequential inference procedures—greedy conditioning and beam search—extend directly to the multi-valued case. For branching and node selection, we can perform greedy selection over variable–value pairs at each node in the branch-and-bound search tree.
>
> Thus, the current architecture extends naturally to multi-valued discrete variables without requiring significant changes to the data collection, model design, or inference procedures.
>
> 2. **Extension to continuous-valued variables:**
>
> **Data Collection:**
> * Direct application is infeasible because the L2C framework operates on variable–value pairs. For continuous domains, discretize each variable into a finite set of representative values (e.g., via uniform binning, quantiles, or problem-specific grid points).
> * During data collection, condition on these discretized values, and collect solver statistics as in the discrete case. To collect better quality data we can employ adaptive binning—allocating more candidate values where the density or utility gradient is high.
>
> **Training:**
> * Initialize embeddings for each (variable, discretized value) pair.
> * The existing attention-based network and both heads (optimality, simplification) apply without architectural change. The pairwise scoring now covers an expanded set.
> * Retain the original objective formulations. Losses apply to target discretized values for optimality and simplification. Soft-label targets can smooth transitions across adjacent bins, improving robustness to discretization artifacts.
>
> **Inference:**
> * Apply greedy or beam search over (variable, discretized value) pairs, as in the multi-valued setting.
> * After selecting promising variables/bins, optionally perform local continuous optimization (e.g., gradient descent, coordinate search) initialized from the best discrete assignments, to refine the solution beyond discretization resolution.
> * Condition or branch on the discretized values;
>
> ## Comment 2
>
> > MAXSAT/PB Solver Comparison: Lacks direct comparison with pseudo-Boolean (PB) solvers, missing an opportunity to contextualize novelty against established methods.
>
> ### Response
>
> Our method is best viewed as a neural conditioning strategy that assists MPE solvers, rather than a standalone solver. Its goal is to improve efficiency and solution quality under resource constraints, especially in settings where exact inference is computationally expensive.
>
> As such, comparisons to heuristic strategies—e.g., graph-based heuristics or strong branching heuristics—are more appropriate. These baselines share the same goal: guiding solvers to improve performance, rather than solving the problem end-to-end.
>
> In Sec. 4.3.1, we evaluate under a fixed time budget, reflecting realistic scenarios such as neurosymbolic inference or time-sensitive applications. As shown in Fig. 2a and 2b, conditioning with our learned heuristic consistently leads to higher-quality solutions within the same time window. Even with generous limits (e.g., 60 seconds), unconditioned solvers often fail to reach comparable quality.
>
> Although pseudo-Boolean (PB) solvers and MAXSAT formulations are relevant for certain inference tasks, a direct comparison is not meaningful in our setup. MPE inference in PGMs involves probabilistic semantics, large factor graphs, and real-valued potentials, which differ fundamentally from the propositional logic and Boolean constraints typically handled by PB solvers. Integrating or benchmarking against PB solvers would require non-trivial translation of the problem and is beyond the scope of this work.

---

> ### Comment · Reviewer_GK5K · 2025-08-05
>
> Thank you for your detailed response. Regarding **Comment 1**, I appreciate your explanation of how the proposed framework could potentially be extended to multi-valued and continuous variables. The ideas seem reasonable and theoretically sound. However, since no implementation or experimental results are provided, I suggest clarifying in the paper that this is a prospective extension rather than a demonstrated capability.
>
> As for **Comment 2**, I would like to point out that encoding MPE inference into PB or weighted MAXSAT is not entirely out of scope. Prior works have explored such transformations—for instance:
>
> * Cussens \[1] shows how Bayesian network learning and inference can be compiled into weighted MAX-SAT;
> * Toulbar2 \[2] supports compiling probabilistic graphical models into WCSP formulations closely related to PB/MaxSAT.
>
> These examples indicate that while a direct comparison may require additional modeling effort, it is not unreasonable and could offer a valuable perspective.
>
> **References:**
>
> \[1] Cussens J. *Bayesian network learning by compiling to weighted MAX-SAT*. arXiv:1206.3244, 2012.
>
> \[2] Toulbar2: [https://github.com/toulbar2/toulbar2](https://github.com/toulbar2/toulbar2)

---

> > ### Author Response · Authors · 2025-08-06
> >
> > Thank you for raising this important point. We agree that additional evaluations with other solver classes, such as pseudo-Boolean or MAXSAT-based solvers, would further strengthen the case for generality. In this work, we have already conducted experiments with two substantially different solver backends: a graphical model-based solver (AND/OR Branch and Bound) and an integer linear programming-based solver (SCIP). These solvers rely on very different internal representations and optimization strategies. Nevertheless, both benefit significantly from the conditioning assignments produced by our learned heuristic.
> >
> > This observation supports the central thesis of our paper:
> > > A neural model can learn to identify variable-value assignments that simplify MPE inference, resulting in substantial reductions in solver runtime and search complexity across diverse solver classes, without compromising solution quality.
> >
> > We anticipate that incorporating additional backends, such as MaxWalkSAT or pseudo-Boolean solvers, will further validate this claim, but are unlikely to change the core insight.
> >
> > We reiterate that the proposed framework is not limited to MPE inference in probabilistic graphical models. It can be extended to other SAT-type and combinatorial optimization tasks, such as MAXSAT, SMT, probabilistic theorem proving, traveling salesman, and facility location, where conditioning on variable assignments can reduce the search space. While these directions are beyond the current scope, the core insight remains the same: a learned model can identify useful assignments that simplify inference and accelerate solver performance. We acknowledge, however, that some of these adaptations may require significant changes, either in terms of implementation or in the development of problem-specific encodings that take into account the structure of the underlying domain (e.g., graphs in TSP, constraints, backbone identification and propagation in MAX-SAT, etc.)

---

> > > ### Comment · Reviewer_GK5K · 2025-08-07
> > >
> > > Thank you for the clarification. I agree that the framework is promising and broadly applicable. However, my concern lies not with the core insight but with the limited empirical validation. The solvers used in your evaluation may not reflect the state of the art, and many modern solvers incorporate strong built-in heuristics that could match or surpass the benefit of your learned assignments. To substantiate your claims, comparisons with more competitive or widely used solvers are necessary.

---

> > > > ### Author Response · Authors · 2025-08-07
> > > >
> > > > Thank you for your thoughtful feedback. We agree that rigorous empirical validation is essential. As far as we are aware, AOBB is considered a state-of-the-art solver with strong empirical performance, including top results in the UAI 2022 competition. We selected it precisely for its competitiveness and relevance to current research.

---

### Official Review · Reviewer_D8Hg · 2025-07-01

**Clarity:** 3
**Significance:** 3
**Originality:** 3
**Rating:** 4
**Confidence:** 3

**Summary:**

This paper introduces L2C, a novel neural method that accelerates MPE inference in PGMs. By learning from exact MPE solvers, L2C identifies strategic variable assignments that effectively balance problem simplification with solution optimality, enabling it to serve as both a powerful pre-processor and an intelligent heuristic within B&B algorithms. Empirical results demonstrate L2C's superior performance over traditional methods, significantly improving the efficiency and tractability of MPE problem-solving.

**Questions:**

See Cons. Specifically:

First question: Is this the first application of Attention to MPE problems?

Second: How significant is the cost of data collection?

Third, this is a question.

Fourth, this is a suggestion.

Fifth, this is a suggestion.

**Ethical Concerns:**

["NO or VERY MINOR ethics concerns only"]

**Final Justification:**

The author answered my questions very well, but considering the differing opinions of other reviewers, and given that my own perspective might not fully encompass every nuance of this specialized field, I believe the current score is appropriate.

**Limitations:**

The authors only discuss limitations in the "Future work" section, specifically stating "Future work will target L2C’s current limitations in scalability, solver signal quality, and the scope of learned guidance." (lines 396), which is insufficient for a comprehensive discussion of shortcomings.

**Paper Formatting Concerns:**

No major issues found.

**Quality:**

3

**Strengths And Weaknesses:**

# Pros:

1.The method is technically sound, features a cleverly designed training objective, and is strongly supported by comprehensive empirical results.

2.L2C effectively addresses a critical challenge in MPE inference, making previously intractable problems solvable.

3.The approach of learning conditioning strategies by combining deep learning with classical MPE solvers is highly novel.

# Cons:
1.If this is the first application of Attention to MPE problems, it would be very significant. However, I'd like to ask, how significant is the difference between L2C and the attention mechanisms in classical Graph Neural Networks?

2.Reliance on computationally expensive 'oracle' solvers might limit its application to large-scale problems. Could we try a Bootstrap method to accelerate data collection?

3.The core insight of the paper is: "A key insight of our method is that solution ambiguity modulates conditioning risk." This statement is somewhat straightforward. It's quite intuitive, much like "high risk, high reward". Could you elaborate further?

4.A more in-depth explanation of the specific information and contextual relationships captured within the learned variable-value pair embeddings would improve clarity. As I am not an expert in this field, it took me a considerable amount of time to understand the specific mechanism of the embeddings.

5.The paper's discussion on L2C's applicability or extensions to other inference types or continuous variables is limited.

---

> ### Author Rebuttal · Authors · 2025-07-30
>
> We thank Reviewer D8Hg for their detailed review and insightful suggestions. Below, we address each comment individually.
>
> > 1. If this is the first application of Attention to MPE problems, it would be very significant. However, I'd like to ask, how significant is the difference between L2C and the attention mechanisms in classical Graph Neural Networks?
>
> To the best of our knowledge, this is the first application of **attention-based models** to the MPE problem in PGMs. While prior work has explored hypergraph neural networks for MPE \[1\], and graph attention networks (GATs) for combinatorial optimization problems such as TSP \[2\], these approaches differ fundamentally from ours.
>
> Our method treats variable–value pairs as learnable embeddings, allowing the multi-head attention mechanism to operate over these pairs directly. This design enables the model to capture fine-grained, context-dependent interactions between query and evidence variables in a high-dimensional space—going beyond standard GAT-style message passing over PGM graphs. In a way, our approach introduces a **novel embedding scheme** that captures information about variables and their assigned values, tailored specifically to the MPE task.
>
> Thus, while our architecture incorporates attention, it is conceptually distinct from classical GNNs and tailored to the structure of the MPE inference problem.
>
> > 2. Reliance on computationally expensive 'oracle' solvers might limit its application to large-scale problems. Could we try a Bootstrap method to accelerate data collection? How significant is the cost of data collection?
>
> During data collection, we solve each unconditioned MPE query exactly only once. As the size of the conditioning set $C$ increases (Algorithm 1), the resulting queries become easier to solve, since fewer variables remain in the query set.
>
> We also incorporate a method to accelerate data collection. Rather than allowing the oracle to run indefinitely after conditioning, we limit execution to 60 seconds and use the observed improvement in log-likelihood, solving time, or number of nodes as the training signal. This enables efficient generation of high-quality supervision without full exact inference.
>
> That said, overly noisy or weak training signals can degrade the learned heuristic's effectiveness, potentially outweighing the benefit of avoiding a few exact queries.
>
>
> > 3. The core insight of the paper is: "A key insight of our method is that solution ambiguity modulates conditioning risk." This statement is somewhat straightforward. It's quite intuitive, much like "high risk, high reward". Could you elaborate further?
>
> Conditioning on an incorrect variable–value pair can prevent recovery of the true MPE solution. This risk is context-dependent: it is highest when a variable consistently takes the same value across optimal solutions, since an incorrect assignment eliminates all optimal paths. Conversely, when a variable's assignment varies across high-probability solutions, even suboptimal choices may still preserve access to near-optimal outcomes.
>
> Our model, trained on diverse solver traces, implicitly learns this trade-off. It assigns higher scores to variable–value pairs where the expected utility outweighs the risk, and lower scores when ambiguity makes early conditioning more likely to prune optimal paths. This learned behavior captures the underlying structure of solution ambiguity and its impact on conditioning risk.
>
>
> > 4. A more in-depth explanation of the specific information and contextual relationships captured within the learned variable-value pair embeddings would improve clarity. As I am not an expert in this field, it took me a considerable amount of time to understand the specific mechanism of the embeddings.
>
> Thank you for pointing this out. We will clarify this in the revised version. Below is a summary of the key design choices:
>
> Each variable–value pair is encoded as a **learnable high-dimensional embedding**, rather than using separate or one-hot encodings for variables and values. This joint representation is critical, as the model must assess the plausibility of specific assignments under a fixed evidence set. Encoding variables and values independently would fail to capture their joint semantics, particularly in high-treewidth PGMs where dependencies are complex.
>
> The **multi-head attention mechanism** then processes these variable–value pair embeddings, enabling the model to reason over their contextual relationships. Attention allows the network to model interactions conditioned on the current evidence, effectively identifying which assignments are informative for conditioning.
>
> This design improves both the expressiveness of the model and its ability to generalize across queries, facilitating accurate and efficient inference.
>
> > 5. The paper's discussion on L2C's applicability or extensions to other inference types or continuous variables is limited.
>
> Our proposed L2C method can be easily extended to multi-valued discrete variables and, with additional modifications, to continuous variables. We provide more details below.
>
> 1. **Extension to multi-valued discrete variables:**
>
> * **Data Collection:** The procedure remains unchanged.
>
> * **Training:** As in the binary case, we initialize embeddings for all variable–value pairs. Our attention-based architecture naturally supports multi-valued variables, as it outputs scores for each pair independently. The optimality and simplification heads produce per-pair scores, and the existing loss functions remain applicable without modification.
>
> * **Inference:** The same sequential inference procedures—greedy conditioning and beam search—extend directly to the multi-valued case. For branching and node selection, we can perform greedy selection over variable–value pairs at each node in the branch-and-bound search tree.
>
> L2C **naturally supports multi-valued variables** with no major changes to data, model, or inference.
>
> 2. **Continuous variables:**
>
> * **Data Collection:** Discretize variables (e.g., binning, quantiles), then collect solver stats on these values. Adaptive binning improves quality.
> * **Training:** Use embeddings for (variable, discretized value) pairs. The model and losses remain unchanged; soft labels improve robustness.
> * **Inference:** Perform greedy or beam search over discretized pairs. Optionally refine with local optimization (e.g., gradient descent) from top assignments.
>
> L2C handles continuous variables via discretization with **minimal architectural changes**.
>
> > 6. The authors only discuss limitations in the "Future work" section, specifically stating "Future work will target L2C’s current limitations in scalability, solver signal quality, and the scope of learned guidance." (lines 396), which is insufficient for a comprehensive discussion of shortcomings.
>
> We thank the reviewer for the suggestion. While the sentence on line 396 introduces the limitations, we do provide further detail in the subsequent lines (397–401), outlining challenges related to scalability, solver signal quality, and the scope of learned guidance:
>
> > “We will adapt L2C for larger models (millions of variables and factors), necessitating efficient large-scale data collection. Richer solver signals (e.g., branch-and-cut decisions) will be integrated for more aggressive pruning, improving upon current signal usage. Finally, learned variable orderings will be explored for novel neural bounding strategies, extending guidance beyond conditioning to further boost inference performance.”
>
> That said, we agree that this discussion remains brief. In the revised version, we will make the limitations more explicit and include them in a dedicated subsection. In particular, we will elaborate on:
>
> * **Scalability:** Data collection and training become increasingly expensive for large PGMs. Further work is needed to reduce computational cost, possibly through model compression, amortized inference, or subgraph-based training.
>
> * **Generalization:** The current method is tailored to fixed PGMs. Extending to unseen or structurally similar PGMs—e.g., via meta-learning or transfer learning—remains an open challenge.
>
> * **Solver dependence:** Training relies on solver-generated signals, which can be expensive to obtain. Although we use proxy signals via time-limited runs, minimizing dependence on oracle solvers is an important future direction.
>
> We will include these limitations in the revised manuscript to provide a more thorough and transparent discussion.
>
> References:
>
> [1]	S. Arya, T. Rahman, and V. G. Gogate, “SINE: Scalable MPE inference for probabilistic graphical models using advanced neural embeddings,” presented at the The 28th International Conference on Artificial Intelligence and Statistics, Feb. 2025.
>
> [2] Luo, J., Heng, H. and Wu, G. Graph attention, learning 2-opt algorithm for the traveling salesman problem. Complex Intell. Syst. 11, 117 (2025). https://doi.org/10.1007/s40747-024-01716-5

---

> > ### Comment · Reviewer_D8Hg · 2025-08-05
> >
> > Thanks for your rebuttal. Since all my concerns have been addressed, I will maintain my positive rating.

---

### Official Review · Reviewer_CV4Q · 2025-07-02

**Clarity:** 3
**Significance:** 2
**Originality:** 2
**Rating:** 2
**Confidence:** 5

**Summary:**

The paper presents a new data-driven framework called Learning to Condition (L2C) for tackling the Most Probable Explanation (MPE) inference problem in probabilistic graphical models. The MPE task in graphical models is known to be computationally intractable. Specifically, the proposed L2C method trains a neural network to score variable-value assignments based on their utility for conditioning. Essentially, L2C can generate cutset conditioning sets that once instantiated are likely to yield a much simplified MPE instance that can be solved efficiently by existing MPE solvers. In addition, the L2C approach can be used as bot variable ordering as well as value ordering heuristics for specialised search-based solvers for MPE such as MIP solvers or AND/OR search based solvers. The experimental evaluation is carried out on standard graphical models benchmarks. The results demonstrate the benefits of the proposed method.

**Questions:**

- Regarding transferability, it would be interesting to train L2C on a given distribution (e.g., such as the one derived from the BN/Grid instances) and experiment with problem instances from a different distribution such as the pedigree or protein networks from the UAI Challenge Benchmark. What would be the performance gap in that case?

- L2C looks very similar to the learning to branch approaches developed recently for mixed integer programming (see ref [31]). What is the conceptual difference between the two approaches?

**Ethical Concerns:**

["NO or VERY MINOR ethics concerns only"]

**Limitations:**

The proposed L2C method appears to be limited to binary variables.

**Paper Formatting Concerns:**

No concerns

**Quality:**

3

**Strengths And Weaknesses:**

Strengths
---------

The MPE (or MAP) inference task in graphical models is a fundamental yet very difficult task with a wide range of real-world applications. Therefore, developing new methods to solving it more efficiently in practice is definitely called for.

The paper is fairly well written and organised. Therefore, the content is relatively easy to follow. The quality of the presentation is overall quite good.

The empirical evaluation, although fairly limited, is sound and the results are presented in a clear manner which makes it relatively easy for the reader to appreciate the benefits of the proposed method.


Weaknesses
----------

The empirical evaluation appears to be limited to a few classes of problems from the UAI Inference Challenge Benchmark (the BN and Grid instance are actually random instances) and the training dataset seems to be generated from the same distribution. Therefore, the performance gains highlighted in the experiments aren't actually surprising. The problem instances shown in Figure 2, although they have relatively high tree widths, are actually fairly easy for exact MPE/MAP inference.

State of the art branch and bound algorithms for MPE/MAP exploit both variable as well as value ordering heuristics. Value ordering heuristics make a huge difference in the case of non-binary domains. However, the benchmarks considered in the experiments (as well as the presentation of the L2C methodology) only considers the case of binary variables.

The experimental evaluation ignores two classes of MPE algorithms: the local search MPE algorithms by Hutter et al, IJCAI-2005 and the exact soft arc-consistency based search algorithms from the toolbar2 suite. The latter use dynamic variable and value ordering heuristics and L2C would be more relevant for them instead of AOBB. The AOBB version considered in this paper, although quite powerful, is limited to a static variable ordering given by the pseudo-tree.

Finally, the MIP formulation for MPE/MAP is known to be not very effective. MIP solvers are typically outperformed by dedicated solvers such as as toolbar and AOBB.

---

> ### Author Rebuttal · Authors · 2025-07-30
>
> We thank Reviewer CV4Q for their valuable comments and suggestions.
>
> >1.1 The evaluation uses only a few random BN and Grid instances from the UAI benchmark.
>
> We selected the BN and Grid instances to introduce structural diversity across different PGMs. In addition, we evaluated our approach on the Promedas class, which contains real-world networks with varying complexity.
>
> This selection aligns with prior work on MPE inference over PGMs \[1,2\], which also used models from the UAI Inference Challenge. The benchmark offers PGMs with diverse treewidths, variable counts, and factor sizes, allowing us to demonstrate that our learned heuristics are **scalable and effective across structurally varied models**.
>
> > 1.2 The training and test data share the same distribution, so the performance gains are unsurprising.
>
> The training and test queries are sampled from the same PGM; however, the model must still generalize over a combinatorially large space of queries, each inducing a distinct inference subproblem.
>
> General-purpose heuristics (e.g., graph-based) are not tailored to specific PGMs and thus often underperform. In contrast, our neural heuristic learns **model-specific structural and parametric patterns**, enabling improved performance across diverse queries, as demonstrated in our experiments.
>
> While this work focuses on the fixed-PGM setting, extending it to support transfer across structurally similar PGMs (e.g., larger Ising models or parameter variations) is an important direction for future work. This would require changes to the model architecture, input representations, and training procedure to capture transferable patterns. This will be the holy-grail, yielding a universally transferable MPE engine that works across model families. If we solve this, we will have achieved an chatgpt‑like breakthrough for optimization, a task that is anything but easy. We like your thought process; you are clearly thinking what the overarching goal should look like.
>
> Our setup is consistent with recent work on neural inference in PGMs \[1\], where a separate network is trained per model to learn its specific structure. In contrast, general-purpose learned heuristics require substantially more data, computational resources, and model capacity, often resulting in degraded performance. Our objective is to develop **PGM-specific neural heuristics** that consistently outperform generic methods in both accuracy and efficiency.
>
> >1.3 The instances in Figure 2 have high treewidth but are easy for exact MPE/MAP inference.
>
> Our method is best viewed as a **neural conditioning strategy that assists MPE solvers**, rather than an MPE solver itself. The goal is to improve solution quality and efficiency under resource constraints, across different PGMs.
>
> In Sec. 4.3.1, we evaluate under a fixed time budget, simulating real-time inference. As Fig. 2a–b show, conditioning yields better solutions within the same limit. Even at 60 seconds, it outperforms the unconditioned solver, highlighting the challenge of exact inference under time constraints.
>
> Furthermore, when exact solutions are obtainable (Fig. 3), our learned heuristic significantly reduces the number of nodes explored while producing near-optimal results. This leads to faster inference and improved solver efficiency.
>
> >2 State-of-the-art BnB algorithms use variable and value ordering, but the experiments and L2C method only address binary variables, missing the impact of value ordering in non-binary cases.
>
> Our approach incorporates both **variable and value ordering** heuristics. Specifically, the model outputs **optimality and ranking scores for variable–value pairs**, not just variables. During training, we initialize separate learnable embeddings for each variable and each of its possible values. The attention-based neural network uses these embeddings to score variable–value pairs, enabling informed decisions during inference.
>
> The method naturally extends to **multi-valued variables**: the number of embeddings and output nodes scales with the domain size, while the rest of the architecture remains unchanged.
>
> Although the current benchmarks involve binary variables, the methodology is fully compatible with non-binary domains. We plan to include experiments on multi-valued PGMs in the camera ready version to demonstrate this capability.
>
> >3 The evaluation omits local search (Hutter et al., 2005) and exact arc-consistency-based methods (toolbar2), which use dynamic heuristics where L2C would be more relevant than the static-order AOBB used.
>
> We appreciate the reviewer's suggestion to explore local search MPE algorithms and soft arc-consistency methods[3], which align well with directions for future work.
>
> This paper focuses on branch-and-bound (B\&B) solvers—both classical (SCIP) and structure-aware (AOBB)—as they dominate exact inference for high-treewidth models and offer natural integration points for learned variable–value scores.
>
> We agree L2C can extend to other solver families:
>
> * **Local Search:** L2C can precondition the problem by fixing high-confidence variable–value pairs, reducing dimensionality and guiding search, though interaction tuning is required.
>
> * **Arc-Consistency [3]:** L2C's variable–value scores can enhance or replace heuristics for adaptive ordering, improving pruning efficiency.
>
> That said, we emphasize that **local search** and **arc-consistency–based solvers** differ fundamentally from branch-and-bound solvers in both mechanism and behavior. While the current L2C-based neural heuristic can be applied to these solvers, its performance may be suboptimal.
>
> Achieving strong results in these settings would require significant changes to L2C-including new data generation, new architecture, and potentially new training objectives. We consider this as promising future work.
>
> Thank you for pointing out the relevant papers and search techniques. We will include the suggested references in the revised version of the paper.
>
> >4 MIP formulations for MPE/MAP are generally ineffective and are outperformed by dedicated solvers like toolbar and AOBB.
>
> Our method is solver-agnostic and serves as a general-purpose conditioning strategy that can be integrated with any systematic solver
>
> We use the MIP solver solely during data collection to generate training labels. This choice is not essential—any systematic search solver, such as AOBB, can be used in its place without affecting the core framework.
>
> Once conditioned via the learned heuristic, the modified query can be solved by any MPE solver. As shown in Sec. 4.3.1, we use the same heuristics with both MIP and AOBB, isolating the effect of conditioning. This confirms the method's **solver-agnostic** nature and broad applicability
>
> >5 To assess transferability, L2C could be trained on BN/Grid instances and tested on different distributions (e.g., pedigree or protein networks) to evaluate the resulting performance gap.
>
> Transfer across PGM classes is important and part of our future work.
>
> **Intra-class transfer** (e.g., small to large Ising models) is more feasible due to shared structure but still requires changes to architecture, embeddings, and training.
>
> **Cross-class transfer** (e.g., BN to Pedigree) is harder due to differences in topology, semantics, and factors. It would need larger models, more training data, and may still degrade performance.
>
> Traditional heuristics avoid this issue by deferring computation to test time. In contrast, our training-intensive approach enables fast, model-specific inference, essential for real-time use.
>
> Our focus is on **PGM-specific neural heuristics** that outperform generic methods in accuracy and efficiency.
>
> >6 L2C closely resembles recent learning-to-branch methods for mixed integer programming (e.g., \[31]); what is the key conceptual difference between them?
>
> While both guide combinatorial solvers via learning, L2C and learning-to-branch differ in key ways:
>
> * **Purpose**: Learning-to-branch selects variables during BnB search; L2C selects variables to condition on before solving, simplifying the problem.
> * **Training signal**: Branching uses imitation learning from strong branching scores; L2C uses supervised learning from solver statistics, without needing optimal traces.
> * **Generalization**: L2C is solver-agnostic—usable for conditioning, branching, or with various solvers (e.g., MIP, AOBB), as shown in Sec. 4.3.
>
> L2C is a broader framework for query-dependent conditioning, not tied to solver internals.
>
> >7 The proposed L2C method appears to be limited to binary variables.
>
> Our proposed L2C method can be easily extended to multi-valued discrete variables and, with additional modifications, to continuous variables. We provide more details below.
>
> * **Multi-valued:**
>   Training uses embeddings for variable–value pairs; the architecture and losses remain unchanged. Inference (greedy, beam, BnB) operates directly over these pairs.
>
> * **Continuous:**
>   Discretize variables (e.g., binning, quantiles). Train and infer over (variable, discretized value) pairs. Optionally refine solutions via local optimization.
>
> These extensions retain the core L2C architecture with minimal changes.
>
> #### References
> [1]	S. Arya, T. Rahman, and V. G. Gogate, “A neural network approach for efficiently answering most probable explanation queries in probabilistic models,” presented at the The Thirty-eighth Annual Conference on Neural Information Processing Systems, Nov. 2024.
>
> [2]	S. Agarwal, K. Kask, A. Ihler, and R. Dechter, “NeuroBE: Escalating neural network approximations of bucket elimination”.
>
> [3] Larrosa, J., & Schiex, T. (2003). In the quest of the best form of local consistency for weighted CSP. Proceedings of the 18th International Joint Conference on Artificial Intelligence, 239–244. Presented at the Acapulco, Mexico. San Francisco, CA, USA: Morgan Kaufmann Publishers Inc.

---

### Official Review · Reviewer_992G · 2025-07-02

**Clarity:** 4
**Significance:** 4
**Originality:** 4
**Rating:** 5
**Confidence:** 4

**Summary:**

This work aims to scale the MPE inference by using a neural network-based conditioning algorithm. It includes first constructing the learning data by collecting subset variable assignments along with the computational costs. Then it uses these data to train a neural network that predicts 1) the likelihood of variable assignments being part of the MPE solution and 2) how efficient the candidate assignments are. At inference time, these scores are combined to select conditioning assignments. Experimental results on some PGMs are presented along with a comparison with a graph-based heuristic baseline and a full strong branching heuristic baseline.

**Questions:**

In addition to those raised in [Strengths And Weaknesses]:
- How well does the learned heuristic perform for out-of-distribution data?
- Many hyperparameters are involved: sampling parameter in the data collection process, the weights in the loss function, size of the conditioning set, etc. How are these hyperparameters chose,n and how sensitive is the performance to these hyperparameters?

**Ethical Concerns:**

["NO or VERY MINOR ethics concerns only"]

**Final Justification:**

The authors have addressed my concerns. Thus I keep my positive score.

**Limitations:**

Yes.

**Paper Formatting Concerns:**

None.

**Quality:**

4

**Strengths And Weaknesses:**

The problem of choosing a subset to condition as a way to speed up MPE inference on PGMs is important and challenging. The learning based approach in this work is novel to me. This paper is overall well written, with a thorough discussion on previous work, proposed methods clearly elaborated, and sufficient background for the readers to understand this work. Still, I have a few concerns:
- It is unclear to me how the PGMs in the data collection process are generated. It seems to me that the learned heuristic only works well when the PGM at inference is similar to those in the data collection process; for example, if the PGMs for training data have relatively low treewidth and the PGM at inference time has high treewidth, very likely the learned heuristic won't perform well since this is out of distribution data.
- The other thing unclear to me is how flexible the learned heuristics are. Can it be applied to various cases when the sizes of the conditioning sets are different, or does it require retraining for each case?
- I think this work will benefit from having some toy example where the optimal MPE solution is known, and comparing how accurate the predictions from the learned heuristics are.

---

> ### Author Rebuttal · Authors · 2025-07-30
>
> We thank Reviewer 992G for their thoughtful and constructive feedback. Below, we address each comment individually.
>
> ## Comment 1
>
> > It is unclear to me how the PGMs in the data collection process are generated. It seems to me that the learned heuristic only works well when the PGM at inference is similar to those in the data collection process; for example, if the PGMs for training data have relatively low treewidth and the PGM at inference time has high treewidth, very likely the learned heuristic won't perform well since this is out of distribution data.
>
> ### Response
>
> All PGMs in the paper are sourced from the UAI 2022 benchmark and not synthetically generated. For each PGM, we train a dedicated learned heuristic using queries specific to that graph. At test time, this heuristic is evaluated on unseen queries from the same PGM. This setup assumes the **PGM structure and parameters remain fixed**, aligning with practical scenarios where inference is repeatedly performed on a known model. Think of the neural network as a MPE compiler: it helps you solve any MPE task over a given generative model more accurately and in less time.
>
> We agree that applying a learned heuristic trained on one PGM to another with significantly different structure is unlikely to generalize. This is by design: our method targets the model-specific query inference setting, where the space of possible queries (variable subsets and evidence configurations) is combinatorially large, even for a single PGM.
>
> General-purpose heuristics (e.g., graph-based) lack this model-specific specialization, which often leads to suboptimal performance, as demonstrated in our experimental results (Sec. 4.3). In contrast, our learned heuristic captures query-specific patterns for a given PGM, enabling substantially improved performance on unseen queries from the same model.
>
> We will highlight these details in the experimental section to ensure the setup and assumptions are clearly stated.
>
> ## Comment 2
>
> > The other thing unclear to me is how flexible the learned heuristics are. Can it be applied to various cases when the sizes of the conditioning sets are different, or does it require retraining for each case?
>
> ### Response
>
> Our model is inherently flexible to varying conditioning set sizes. During inference, conditioning on a variable reduces the query set size by one and increases the evidence set size by one. Consequently, even within a single query, the heuristic must operate across a range of evidence sizes.
>
> The data collection process accounts for this variability. As shown in Algorithm 1 (last two for-loops), we generate training data over diverse evidence sizes, enabling the heuristic to generalize across different conditioning set sizes. Therefore, **we do not retrain the model for each conditioning size**. A single neural heuristic is trained per PGM for L2C.
>
> In Sec. 4.3.1, we report results for four conditioning set sizes: 5\%, 10\%, 15\%, and 25\% of variables. Importantly, **a single neural heuristic** handles scoring across all sizes, ranging from 1 up to the number of variables corresponding to these percentages. Our method consistently outperforms baselines by a significant margin across all settings.
>
> Further, as demonstrated in Sec. 4.3.2, the trained heuristic integrates directly into the branch-and-bound solver as both a branching and node selection strategy. It operates effectively across search tree levels where partial evidence of varying sizes is encountered.
>
>
> ## Comment 3
>
> > I think this work will benefit from having some toy example where the optimal MPE solution is known, and comparing how accurate the predictions from the learned heuristics are.
>
> ### Response
>
> Thank you for your thoughtful comment. We completely agree that comparing against true solutions is essential for a thorough evaluation. In fact, this analysis is **already included in the paper**.
>
> As described in Sec. 4.3.1 (“*Assisting AOBB via conditioning*”) and detailed further in the supplemental material, we use **AOBB as an oracle** to obtain exact MPE solutions.
>
> In these experiments, we run AOBB to completion—without time limits—to obtain the **optimal MPE solution** for each instance (see Footnote 2). This ensures that the unconditioned AOBB results serve as the ground truth for optimality.
>
> To assess the accuracy of the learned heuristic, we report the *Avg \% Gap* between the MPE value obtained by the conditioned solver and the optimal unconditioned solution (see Fig. 3). A smaller gap indicates a solution closer to the true optimum. Our method consistently achieves low gap values, outperforming baseline heuristics across multiple conditioning sizes.
>
> This evaluation effectively demonstrates that the learned heuristic yields **near-optimal solutions** while substantially reducing **computational effort**, as measured by the **number of nodes explored** (see y-axis).
>
> ## Comment 4
>
> > How well does the learned heuristic perform for out-of-distribution data?
>
> ### Response
>
> We acknowledge the importance of evaluating the learned heuristic on out-of-distribution PGMs. While this work focuses on the fixed-PGM setting, investigating **transferability to unseen but structurally similar PGMs** (e.g., larger Ising models or models with the same graph structure but different parameters) is a promising direction with practical relevance.
>
> Such generalization would require modifications to the architecture, embeddings, and training procedure to capture transferable patterns across related PGMs. We leave this as future work.
>
> That said, transferring heuristics across **unrelated PGMs** is inherently challenging. Each PGM defines a distinct distribution over a different variable set and structure, making it unlikely that a heuristic trained on one model would generalize well to another without significant adaptation.
>
>
> ## Comment 5
>
> > Many hyperparameters are involved: sampling parameter in the data collection process, the weights in the loss function, size of the conditioning set, etc. How are these hyperparameters chosen and how sensitive is the performance to these hyperparameters?
>
> The **sampling parameter** in the data collection process controls the number of conditioning variables sampled per query. Larger values yield a richer dataset and higher-quality heuristics but incur greater computational cost. We set this parameter based on available computational resources.
>
> The **loss function weights** were selected using a validation set. We performed a grid search over $\lambda\_{\text{opt}} \in \\{0.3, 0.35, 0.4, 0.45, 0.5\\}$ with $\lambda\_{\text{rank}} = 1 - \lambda\_{\text{opt}}$.
>
> The **conditioning set size** represents a trade-off between solution quality and computational efficiency. Larger conditioning sets simplify the remaining inference problem, often improving runtime, but can reduce solution quality. We evaluated sizes from 5\% to 25\% of the total variables to cover this trade-off space. As expected, increasing the conditioning size generally improves efficiency but may slightly degrade optimality.
>
> We include experiments with these varying conditioning set sizes to reflect scenarios where inference must be performed under **strict time constraints**, such as in neurosymbolic systems requiring real-time responses. In these cases, conditioning enables faster convergence to high-quality solutions.
>
> As shown in the SCIP experiments (Sec. 4.3.1), unconditioned inference often fails to reach optimality within the time budget, whereas increasing the conditioning set size yields better log-likelihood (LL) values. This illustrates a **practical trade-off**: although larger conditioning sets may slightly reduce optimality in some cases, they allow the solver to find **better approximate solutions within limited time**.

---

> > ### Comment · Reviewer_992G · 2025-08-05
> >
> > Thanks for your reply. I will keep my positive score.

---

### Author Response · Authors · 2025-08-04
**General response to all the reviewers**

We thank the reviewers for their insightful comments and careful evaluation of our work; we appreciate the constructive feedback, which has enabled us to clarify our contributions, address key concerns, and further strengthen the manuscript in response to your suggestions.

We are encouraged by the recognition of our work’s **significance**, **novelty**, and **potential impact**:

* *“The problem of choosing a subset to condition as a way to speed up MPE inference on PGMs is important and challenging. The learning-based approach in this work is novel to me.”* — **Reviewer 992G**
* *“The method is technically sound, features a cleverly designed training objective, and is strongly supported by comprehensive empirical results.”* — **Reviewer D8Hg**
* *“The approach of learning conditioning strategies by combining deep learning with classical MPE solvers is highly novel.”* — **Reviewer D8Hg**
* *“Combines neural networks with symbolic reasoning, using solver-guided feedback to ensure semantic correctness of generated constraints.”* — **Reviewer GK5K**
* *“The supervision strategy leverages solver statistics without requiring exhaustive MPE enumeration.”* — **Reviewer GK5K**
* *“The paper is fairly well written and organised. Therefore, the content is relatively easy to follow. The quality of the presentation is overall quite good.”* — **Reviewer CV4Q**

We now address common concerns raised across reviews. More detailed responses are provided individually.

* **Applicability beyond binary variables:**
  While our experiments focus on binary PGMs, the architecture extends naturally to multi-valued discrete variables. The attention-based mechanism scores arbitrary variable–value pairs, and the training losses and inference procedures remain unchanged. For continuous variables, discretization (e.g., via uniform or adaptive binning) enables compatibility with the current framework. This can be further refined through local search to improve solution quality beyond discretization granularity.

* **Out-of-distribution generalization:**
  Transferability to unseen PGMs is an important direction and part of our future work. Intra-class generalization (e.g., from smaller to larger Ising models or from different parameterizations of the same structure) is promising but requires adaptation of the embedding scheme and training procedure. Cross-class generalization (e.g., from BNs to protein networks) is significantly more challenging due to differences in topology, semantics, and variable interactions. Our model is training-intensive by design, optimized for fast inference on a fixed PGM, where it consistently outperforms general-purpose heuristics in both quality and efficiency. This setup is valuable in scenarios demanding repeated inference on a known model, such as neurosymbolic pipelines or online decision systems.

---

### Note · Authors · 2025-08-13

We thank the reviewers for their constructive feedback and hope that we have addressed all their concerns.

**Beyond binary variables** Our L2C extends directly to multi-valued discrete variables with unchanged data collection, architecture, losses, and inference. For continuous variables, we can discretize the variables into bins, train on (variable, bin) pairs, and optionally refine by local continuous optimization from the best bins.

**Solver diversity and competitiveness** Our experiments span two competitive and fundamentally different solver backends—AOBB, a top-performing graphical model BnB solver in the UAI 2022 competition, and SCIP, a state-of-the-art ILP solver. These represent distinct internal architectures and heuristic strategies, yet both benefit significantly from our learned conditioning assignments. While additional evaluations with other solver classes (e.g., pseudo-Boolean, MAXSAT, local search methods, or soft-arc consistency–based solvers) could be informative, integrating such backends often requires significant changes to the architecture, data collection procedures, and learning objectives. The current choice of backends was made to ensure strong, representative baselines while keeping the scope tractable.

**Empirical scope** Our aim is complementarity to strong built-in heuristics. Even when native heuristics are competitive, L2C often yields gains by proposing evidence that simplifies subsequent search.

**Flexibility of learned heuristics.** A single model per PGM is trained over diverse evidence sizes and used across varying conditioning set without retraining; the same network also drives branching and node selection inside BnB.

**Out-of-distribution generalization** Intra-class transfer (e.g., larger Ising or parameter shifts) is promising with embedding/augmentation tweaks; cross-class transfer (e.g., BNs → protein networks) is harder. Our design targets repeated inference on a fixed PGM, where we consistently outperform general-purpose heuristics. Extending L2C to improve OOD transfer is an important avenue for future work.

**Data collection cost.** Each base (unconditioned) query is solved exactly once; conditioned queries become easier. We cap oracle runs at 60s and train from partial signals (Δ log-likelihood/time/nodes), which yields high-quality supervision efficiently.

We appreciate the suggestions and will include these details in the camera-ready version while preserving the central thesis.

---

### Decision · Program_Chairs · 2025-09-17

**Decision:**

Accept (poster)

**Comment:**

This paper introduces Learning to Condition (L2C), a novel framework that trains a neural network to learn heuristics for accelerating Most Probable Explanation (MPE) inference. The model learns to predict high-utility variable assignments for conditioning, effectively simplifying the problem for classical solvers. The reviews were largely positive, with three of the four reviewers leaning towards acceptance after a thorough discussion period. The 4th reviewer, CV4Q, did not participate in the discussion, only thanked the authors for clarifications, without providing any final recommendations or justifications for keeping the reject score.

AC recommends acceptance due the following strengths:

* Novelty and Significance: The paper addresses a challenging and important problem in probabilistic inference with a novel learning-based approach. The integration of deep learning with classical MPE solvers is seen as a creative and impactful contribution (R-992G, R-D8Hg, R-GK5K).

* Technical Soundness: The method is technically solid, featuring a well-designed training objective and a flexible architecture that can serve as both a pre-processing step and an integrated branching heuristic for branch-and-bound solvers (R-D8Hg, R-GK5K).

* Empirical Results: The experiments, conducted on high-treewidth graphical models, convincingly demonstrate that L2C significantly reduces the search space and improves solution quality compared to strong baselines, including competitive solvers from the UAI 2022 competition (R-992G, R-D8Hg).

* Clarity and Responsiveness: The paper is well-written and easy to follow. The authors provided thorough and convincing rebuttals that successfully addressed the majority of the reviewers' concerns regarding the experimental setup, model flexibility, and out-of-distribution performance (R-992G, R-D8Hg, R-GK5K).

While there are valid points about extending the evaluation to other solver classes and non-binary domains, the core contribution is well-defined and supported. The work represents a valuable and promising step in combining neural methods with symbolic reasoning for complex inference tasks.